# ALLEVIATING EXPOSURE BIAS IN DIFFUSION MODELS THROUGH SAMPLING WITH SHIFTED TIME STEPS

**Mingxiao Li**[*1]**, Tingyu Qu**[*1]**, Ruicong Yao**[2]**, Wei Sun**[1] **& Marie-Francine Moens**[1]
[1] Department of Computer Science, KU Leuven
[2] Department of Mathematics, KU Leuven
{mingxiao.li,tingyu.qu,ruicong.yao,sun.wei,sien.moens}@kuleuven.be

## ABSTRACT

Diffusion Probabilistic Models (DPM) have shown remarkable efficacy in the synthesis of high-quality images. However, their inference process characteristically requires numerous, potentially hundreds, of iterative steps, which could exaggerate the problem of exposure bias due to the training and inference discrepancy. Previous work has attempted to mitigate this issue by perturbing inputs during training, which consequently mandates the retraining of the DPM. In this work, we conduct a systematic study of exposure bias in DPM and, intriguingly, we find that the exposure bias could be alleviated with a novel sampling method that we propose, without retraining the model. We empirically and theoretically show that, during inference, for each backward time step $t$ and corresponding state $\hat{x}_t$, there might exist another time step $t_s$ which exhibits superior coupling with $\hat{x}_t$. Based on this finding, we introduce a sampling method named Time-Shift Sampler. Our framework can be seamlessly integrated to existing sampling algorithms, such as DDPM, DDIM and other high-order solvers, inducing merely minimal additional computations. Experimental results show our method brings significant and consistent improvements in FID scores on different datasets and sampling methods. For example, integrating Time-Shift Sampler to F-PNDM yields a FID=3.88, achieving 44.49% improvements as compared to F-PNDM, on CIFAR-10 with 10 sampling steps, which is more performant than the vanilla DDIM with 100 sampling steps. Our code is available at https://github.com/Mingxiao-Li/TS-DPM.

## 1 INTRODUCTION

Diffusion Probabilistic Models (DPMs) (Ho et al., 2020; Sohl-Dickstein et al., 2015) are a class of generative models that has shown great potential in generating high-quality images. (Dhariwal & Nichol, 2021; Ramesh et al., 2022; Rombach et al., 2022a; Nichol et al., 2022). DPM consists of a forward and a backward process. In the forward process, images are progressively corrupted with Gaussian noise in a series of time steps. Conversely, during the backward process, the trained diffusion model generates images by sequentially denoising the white noise.

Despite its success in generating high-quality images, DPMs suffer from the drawbacks of prolonged inference time. Considerable interest has been expressed in minimizing the number of inference steps during the sampling process to speed up generation, while preserving the quality of generated images. Such works include generalizing DDPM (Ho et al., 2020) to non-Markovian processes (Song et al., 2021), deriving optimal variance during sampling (Bao et al., 2022b), thresholding the pixel values as additional regularization (Saharia et al., 2022a), or developing pseudo numerical methods for solving differential equations on manifolds (Liu et al., 2022a).

Though showing promising performance, none of them have theoretically and empirically examined the discrepancy between the training and sampling process of DPM. If we take a closer look at the training of DPM, at each time step $t$, ground truth samples $x_0$ are given to produce corrupted samples $x_t$ with noise $\epsilon_t$. The DPM takes both $x_t$ and $t$ as input to predict the noise $\epsilon_t$. During

---

*Equal Contribution

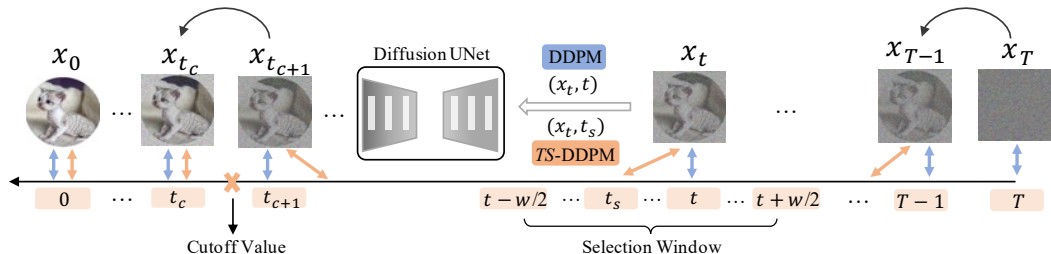

Figure 1: The comparison of *TS*-DDPM (ours) and DDPM. The orange and blue arrows denote the time-state coupling at each denoising step of *TS*-DDPM and DDPM, respectively. In *TS*-DDPM, we search for coupled time step within the $[t - w/2, t + w/2]$ window, until the cutoff time step $t_c$.

sampling, one is required to synthesize data samples from white noise without the knowledge of the ground truth distributions. Coupled with the prediction errors of the network, this training-sampling discrepancy produces errors that could progressively accumulate during inference. This error arising from the difference between training and inference resembles the exposure bias problem as identified in autoregressive generative models (Ranzato et al., 2016; Schmidt, 2019), given that the network is solely trained with the corrupted ground truth samples, rather than the network predicted samples.

In this work, we focus on the exposure bias problem during sampling. Ning et al. (2023) propose to add perturbation to training samples to alleviate the exposure bias problem, which is sub-optimal since the retraining of DPM is computationally expensive. Given that the time step $t$ is directly linked to the corruption level of the data samples, we theoretically and empirically show that by adjusting the next time step $t - 1$ during sampling according to the approximated variance of the current generated samples, one can effectively alleviate the exposure bias. We search for such a time step within a window $t_w$ surrounding the current time step to restrict the denoising progress. Furthermore, based on the error patterns that the network makes on the training samples, we propose the use of a cutoff time step $t_c$. For time steps larger than $t_c$, we search for the suitable time step within $t_w$. While for time steps smaller than $t_c$, we keep the original time step. Intuitively, it also suits the nature of a DPM, since the corruption level is smaller for small time steps. We refer to our sampling method as Time-Shift Sampler. Figure 1 presents the comparison between DDPM, the stochastic sampling method, and its time-shift variant *TS*-DDPM. In summary, our contributions are:

- We theoretically and empirically study the exposure bias problem of diffusion models, which is often neglected by previous works.

- We propose a new sampling method called Time-Shift Sampler to alleviate the exposure bias problem, which avoids retraining the models. Our method can be seamlessly integrated into existing sampling methods by only introducing minimal computational cost.

- Our Time-Shift Sampler shows consistent and significant improvements over various sampling methods on commonly used image generation benchmarks, indicating the effectiveness of our framework. Notably, our method improves the FID score on CIFAR-10 from F-PNDM (Liu et al., 2022a) by 44.49% to 3.88 with only 10 sampling steps.

## 2 INVESTIGATING EXPOSURE BIAS IN DIFFUSION PROBABILISTIC MODELS

In this section, we first give a brief introduction of the training and inference procedure for Diffusion Probabilistic Models (DPMs). Then we empirically study the exposure bias problem in DPM by diving deep into the training and inference processes.

### 2.1 BACKGROUND: DIFFUSION PROBABILISTIC MODELS

DPM encompasses a forward process which induces corruption in a data sample (e.g., an image) via Gaussian noise, and a corresponding inverse process aiming to revert this process in order to generate an image from standard Gaussian noise. Given a data distribution $q(x_0)$ and a forward

noise schedule $\beta_t \in (0,1), t = 1 \cdots T$, the forward process is structured as a Markov process, which can be expressed as:

$$q(x_{1\cdots T}|x_0) = \prod_{t=1}^{T} q(x_t|x_{t-1}) \tag{1}$$

with the transition kernel $q(x_t|x_{t-1}) = \mathcal{N}(x_t|\sqrt{\alpha_t}x_{t-1}, \beta_t\mathbf{I})$, where $\mathbf{I}$ denotes the identity matrix, $\alpha_t$ and $\beta_t$ are scalars and $\alpha_t = 1 - \beta_t$. With the reparameterization trick, the noisy intermediate state $x_t$ can be computed by the equation below:

$$x_t = \sqrt{\overline{\alpha_t}}x_0 + \sqrt{1 - \overline{\alpha_t}}\epsilon_t \tag{2}$$

where $\overline{\alpha}_t = \prod_{i=1}^{t}\alpha_t$ and $\epsilon_t \sim \mathcal{N}(\mathbf{0}, \mathbf{I})$. According to the conditional Gaussian distribution, we have the transition kernel of backward process as:

$$p(x_{t-1}|x_t, x_0) = \mathcal{N}(\tilde{\mu}_t(x_t, x_0), \tilde{\beta}_t) \tag{3}$$

where $\tilde{\beta}_t = \frac{1-\overline{\alpha}_{t-1}}{1-\overline{\alpha}_t}\beta_t$ and $\tilde{\mu}_t = \frac{\sqrt{\overline{\alpha}_{t-1}}\beta_t}{1-\overline{\alpha}_t}x_0 + \frac{\sqrt{\alpha_t}(1-\overline{\alpha}_{t-1})}{1-\overline{\alpha}_t}x_t$. Considering Equation 2, $\tilde{\mu}_t$ can be further reformulated as $\tilde{\mu}_t = \frac{1}{\sqrt{\alpha_t}}(x_t - \frac{1-\alpha_t}{\sqrt{1-\overline{\alpha}_t}}\epsilon_t)$. During training, a time-dependent neural network is optimized by either learning $\tilde{\mu}_t$ or $\epsilon_t$. Empirically, Ho et al. (2020) observe that predicting $\epsilon_t$ works better. The learning target is to optimize the variational lower bound of the negative log-likelihood, which could also be interpreted as minimizing the KL divergence between the forward and backward process. In practice, Ho et al. (2020) further simplify the loss function as:

$$\mathcal{L}_{simple} = \mathbb{E}_{t,x_0,\epsilon_t \sim \mathcal{N}(\mathbf{0},\mathbf{I})}[\|\epsilon_\theta(x_t, t) - \epsilon_t\|_2^2] \tag{4}$$

We present the training and sampling algorithms of the original Denoising Diffusion Probabilistic Models (DDPM) (Ho et al., 2020) in Algorithm 1 and 2, respectively.

---

**Algorithm 1** Training

1: **repeat**
2:      $x_0 \sim q(x_0)$
3:      $t \sim \text{Uniform}(1, \cdots, T)$
4:      $\epsilon \sim \mathcal{N}(\mathbf{0}, \mathbf{I})$ Compute $x_t$ using Eq 2
5:      Take gradient descent step on
6:      $\nabla \|\epsilon - \epsilon_\theta(x_t, t)\|^2$
7: **until** converged

---

**Algorithm 2** Sampling

1: $x_T \sim \mathcal{N}(0, \mathbf{I})$
2: **for** $t = T, \cdots, 1$ **do**
3:      $z \sim \mathcal{N}(\mathbf{0}, \mathbf{I})$ if $t > 1$, else $z = 0$
4:      $x_{t-1} = \frac{1}{\sqrt{\alpha_t}}(x_t - \frac{1-\alpha_t}{\sqrt{1-\overline{\alpha}_t}}\epsilon_\theta(x_t, t)) + \sigma_t z$
5: **end for**
6: **return** $x_0$

---

## 2.2 THE EXPOSURE BIAS IN DIFFUSION PROBABILISTIC MODELS

In this section, we empirically demonstrate the phenomenon related to the exposure bias problem in DPM using CIFAR-10 dataset (Krizhevsky, 2009). We first present the variance distribution of the corrupted samples by different time steps in the forward process during training. Models exposed to a wider range of inputs during training tend to exhibit greater robustness to noise and are consequently less susceptible to exposure bias. To further study the behavior of the network during the backward sampling process, we also examine the evolution of prediction errors during sampling. We use DDIM (Song et al., 2021) sampler to conduct this experiment, as it gives a deterministic sampling process.

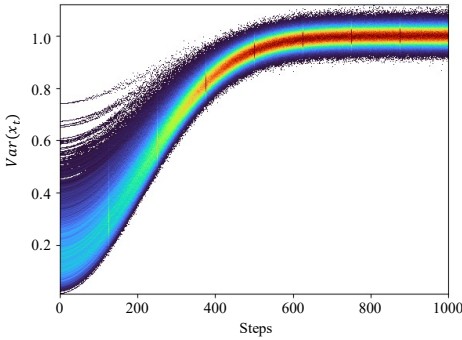

Figure 2: The density distribution of the variance of 5000 samples from CIFAR-10 by different time steps.

Figure 2 presents the changes in variance of sample distributions for different time steps. At each step, we estimate the corrupted image samples with the network predicted noise using Equation 2. We present the details of the figure in Appendix B. At time step 0, ground truth images serve as the current samples. The distribution of the variance of the ground truth samples spans an approximate

range of $(0, 0.8)$, showing the diversity of the sample distributions. As the noise is gradually added to the ground truth samples, the span of the variance becomes narrower and narrower. Following 400 steps, the changes in the span range of the variance become stable, and gradually shift towards a narrow range surrounding the variance of white noise. The evolution of the sample variance across different time steps indicates that the network exhibits a lower sensitivity to the early steps of the forward process of DPM, as the variance of the samples can be distributed more sparsely within a broader range. Conversely, the network can be more sensitive to the later steps (e.g., after 400 steps), as we progressively approach white noise. The constricted variance range during the later stages implies that minor prediction errors during sampling can significantly impact overall performance.

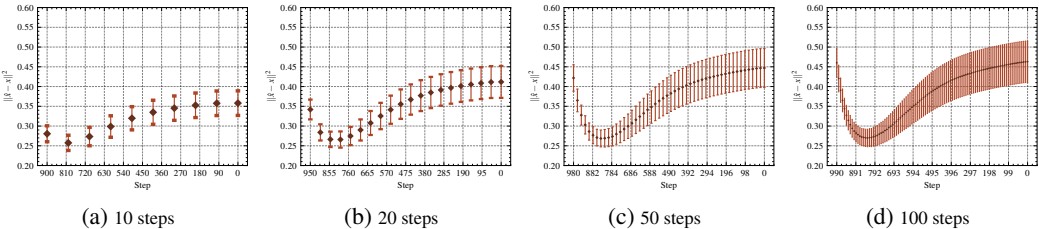

Figure 3: CIFAR-10 prediction errors of training samples for different numbers of sampling steps.

In the second experiment, given a specific number of sampling steps, we compute the mean squared errors between the predicted samples and the ground truth samples at each step, as presented in Figure 3. Details for plotting this figure are presented in Appendix B. It can be seen that the evolution of prediction errors adheres to a consistent pattern: initially decreasing before incrementally accumulating as the sampling process progresses. This phenomenon may be attributed to two possible factors: (1) The sampling process originates from white noise, which does not contain any information about the resultant sample distributions. In early stages, with fewer sampling steps, the error accumulation is less serious, thus the network gradually shapes the predicted distribution into the target distribution. (2) In the later stages, the network is more robust to the noisy inputs as discussed above. However, due to the exposure bias, the network inevitably makes errors at each step and these errors accumulate along the sequence, resulting in a slow but steady progress in error accumulation and larger errors in the end.

In conclusion, the above two experiments together show that during the backward sampling process, the accumulated prediction error, which arises from the exposure bias and the capability of the network, could strongly impact the final results. This demonstrates the importance of alleviating exposure bias in DPM, which could potentially lead to improved results.

## 3 ALLEVIATING EXPOSURE BIAS VIA TIME STEP SHIFTING

In the backward process of Diffusion Probabilistic Models (DPM), the transition kernel is assumed to adhere to a Gaussian distribution. To maintain this assumption, the difference between two successive steps must be sufficiently small, thereby necessitating the extensive training of DPM with hundreds of steps. As previously discussed, the network prediction error coupled with discrepancy between training and inference phases inevitably results in the problem of exposure bias in DPM. We introduce $\mathcal{C}(\tilde{x}_t, t)$–referred to as the input couple for a trained DPM–to describe this discrepancy, which can be expressed as:

$$\mathcal{C}(\tilde{x}_t, t) = e^{-dis(\tilde{x}_t, x_t)} \tag{5}$$

where $\tilde{x}_t$ and $x_t$ represent the network input and ground truth states at time step $t$, respectively, and $dis(\cdot, \cdot)$ denotes the Euclidean distance. Consequently, during the training phase, the relationship $\mathcal{C}(\tilde{x}_t, t) = 1$ holds true for all time steps, as the network always takes ground truth $x_t$ as input. Moreover, a better coupling expressed by $\mathcal{C}(\tilde{x}_t, t_s)$ reduces the discrepancy between training and inference, thereby alleviating exposure bias. Previous works (Zhang et al., 2023a; Ning et al., 2023) have empirically and statistically affirmed that the network prediction error of DPM follows a normal distribution. In conjunction with Equation 2, during the backward process at time step $t$, the

predicted next state denoted as $\hat{x}_{t-1}$ could be represented as:

$$
\begin{aligned}
\hat{x}_{t-1} &= x_{t-1} + \phi_{t-1} e_{t-1} \\
&= \sqrt{\overline{\alpha}_{t-1}} x_0 + \sqrt{1 - \overline{\alpha}_{t-1}} \epsilon_{t-1} + \phi_{t-1} e_{t-1} \\
&= \sqrt{\overline{\alpha}_{t-1}} x_0 + \lambda_{t-1} \tilde{\epsilon}_{t-1}
\end{aligned}
\tag{6}
$$

In this equation, $\lambda_{t-1}^2 = \phi_{t-1}^2 + (1 - \overline{\alpha}_{t-1})$, $x_{t-1}$ denotes the ground truth at time step $t-1$, $\phi_{t-1} e_{t-1}$ represents the network prediction errors, and $e_{t-1}$, $\epsilon_{t-1}$ and $\tilde{\epsilon}_{t-1}$ conform to a normal distribution. Upon observing that Equation 6 and Equation 2 share a similar structure which is the ground truth $x_0$ plus Gaussian noise with variable variance, we propose the subsequent assumption.

**Assumption 3.1** *During inference at time step $t$, the next state $\hat{x}_{t-1}$ predicted by the network, may not optimally align with time step $t-1$ within the context of the pretrained diffusion model. In other words, there might be an alternate time step $t_s$, that potentially couples better with $\hat{x}_{t-1}$:*

$$
\exists t_s \in \{1 \cdots T\}, \quad s.t. \quad \mathcal{C}(\hat{x}_{t-1}, t_s) \geq \mathcal{C}(\hat{x}_{t-1}, t-1)
\tag{7}
$$

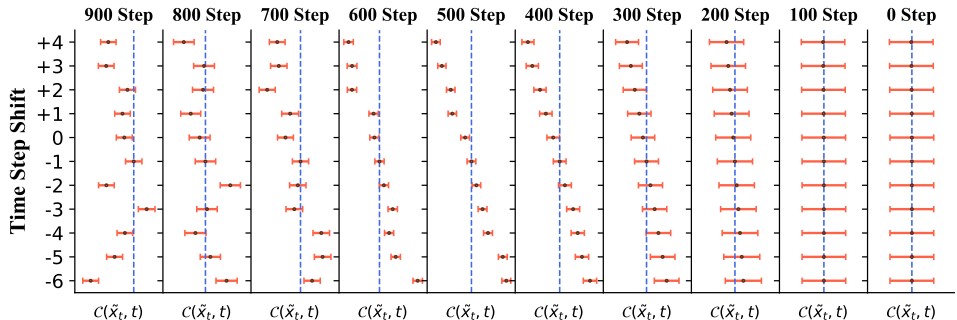

Figure 4: The training and inference discrepancy of DDIM with 10 sampling steps on CIFAR-10. The dashed line in each column denotes the couple of predicted $\hat{x}_t$ and $t$. Points on the right side of the dashed line mean that the corresponding time steps couple better with $\hat{x}_t$ than time step $t$.

To verify our assumption, we initially conduct a statistical examination of the discrepancy between training and inference in a pretrained diffusion model. A sample size of 5000 instances was randomly selected from the CIFAR-10 training set and Equation 2 was utilized to generate ground truth states $x_t$ for a sequence of time steps. Subsequently, we compare the $\mathcal{C}(\hat{x}_t, t)$ with $\mathcal{C}(\hat{x}_t, t_s)$. Details of plotting this figure are presented in Appendix B. Here we show the results of 10 inference steps in the backward process and only consider time step $t_s$ within the range of $t-6$ to $t+4$. As depicted in Figure 4, for certain backward steps, there are alternate time steps $t_s$ that display a stronger correlation with the predicted next state $\hat{x}_t$ compared to time step $t$. We also observe that when approaching the zero time step, all nearby time steps converge to the same distribution. Similar findings were observed for other pretrained DPMs on different datasets including CelebA with varying settings, such as different numbers of inference steps and ranges of $t_s$. See Appendix E for details.

Our empirical findings lend substantial support to Assumption 3.1. This naturally prompts the question: How can we identify the time step that best couples with the predicted $\hat{x}_{t-1}$? By optimizing the KL divergence between the predicted $\hat{x}_{t-1}$ and $x_{t_s}$ at time step $t_s$, we arrive at the following Theorem 3.1, with the complete derivation provided in Appendix J.1.

**Theorem 3.1** *Let $\hat{x}_t$ represent a given state and $\hat{x}_{t-1}$ represent the predicted subsequent state. We assume $t-1$ is sufficiently large such that the distribution of $\hat{x}_{t-1}$ is still close to the initialized normal distribution with a diagonal covariance matrix. In addition, the selected time step $t_s$ to couple with $\hat{x}_{t-1}$ is among those time steps closely surrounding $t-1$. [1] Then the optimal $t_s$ should have the following variance:*

$$
\sigma_{t_s} \approx \sigma_{t-1} - \frac{\|e\|^2}{d(d-1)}
\tag{8}
$$

*where $d$ is the dimension of the input, $e$ represents the network prediction error, and $\sigma_{t-1}$ is the variance of the predicted $\hat{x}_{t-1}$.*

---

[1] If $t_s$ is close to $t-1$, then $\sqrt{\overline{\alpha}_{t-1}} - \sqrt{\overline{\alpha}_{t_s}} \approx 0$. See details in Appendix J.1

The derivation of Theorem 3.1 mainly follows two steps: Firstly, we optimize the KL divergence between $x_{t_s}$ and $\hat{x}_{t-1}$ to obtain the variance of $x_{t_s}$. Secondly, we establish the relationship between the variance within a single sample of $\hat{x}_{t-1}$ and the variance of $\hat{x}_{t-1}$. [2] The results articulated in Theorem 3.1 could be further simplified to $\sigma_{t_s} \approx \sigma_{t-1}$, when $t$ is large and given the assumption that the network prediction error at the current time step is minimal. This assumption has been found to hold well in practice.

Based on the findings of Theorem 3.1, we propose the Time-Shift Sampler, a method that can be seamlessly incorporated into existing sampling algorithms, such as Denoising Diffusion Implicit Models (DDIM) (Song et al., 2021), Denoising Diffusion Probabilistic Models (DDPM) (Ho et al., 2020) or sampling methods based on high-order numerical solvers (Liu et al., 2022a). Moreover, in light of the earlier discussion that the model's predictions of nearby time steps tend to converge to the same distribution during the later stage of the inference process and the condition of large $t$ in the derivation in Appendix J.1, we remove the time-shift operation when the time step is smaller then a predefined cutoff time step.[3] Our algorithm is detailed in Algorithm 3. Specifically, given a trained DPM $\epsilon_\theta$, we sample for $x_t, t = 1, 2, \ldots, T$ using arbitrarily any sampling method. For each time step $t > t_c$, where $t_c$ is the cutoff threshold, we replace the next time step $t_{next}$ with the time step $t_s$ that best couples with the variance of $x_{t-1}$ within a window of size $w$. In the search of such $t_s$, we first take a sampling step with the current $t_{next}$ to get $x_{t-1}$, which is used to compute the variance of $x_{t-1}$ as $var(x_{t-1})$. Then we get the variance $\Sigma$ of the time steps within the window. The optimal $t_s$ can be obtained via $\arg\min_\tau ||var(x_{t-1}), \sigma_\tau||$, for $\sigma_\tau \in \Sigma$. Finally, the obtained $t_s$ is passed to the next sampling iteration as $t_{next}$. We repeat this process until $t < t_c$, after which we perform the conventional sampling steps from the sampling method of choice.

---

**Algorithm 3** Time-Shift Sampler

1: **Input :** Trained diffusion model $\epsilon_\theta$; Window size $w$; Reverse Time series $\{T, T - 1, \cdots, 0\}$; Cutoff threshold $t_c$
2: **Initialize:** $x_T \sim \mathcal{N}(\mathbf{0}, \mathbf{I})$ ; $t_s = -1$
3: **for** $t = T, T - 1, .., 0$ **do**
4:     If $t_s \neq -1$ then $t_{next} = t_s$ else $t_{next} = t$
5:     $\epsilon_t = \epsilon_\theta(x_t, t_{next})$
6:     take a sampling step with $t_{next}$ to get $x_{t-1}$
7:     **if** $t > t_c$ **then**
8:         Get variance for time steps within the window: $\Sigma = \{1 - \overline{\alpha}_{t-w/2}, 1 - \overline{\alpha}_{t-w/2+1}, \cdots, 1 - \overline{\alpha}_{t+w/2}\}$
9:         $t_s = \arg\min_\tau ||var(x_{t-1}) - \sigma_\tau||$, for $\sigma_\tau \in \Sigma$ and $\tau \in [t - w/2, t + w/2]$
10:     **else**
11:         $t_s = -1$
12:     **end if**
13: **end for**
14: **return** $x_0$

---

## 4 EXPERIMENTAL SETUP

We integrate our Time-Shift Samplers to various sampling methods including **DDPM** (Ho et al., 2020): the stochastic sampling method; **DDIM** (Song et al., 2021): the deterministic version of DDPM; **S-PNDM** (Liu et al., 2022a): the sampling method based on second-order ODE solver; and **F-PNDM** (Liu et al., 2022a): the sampling method based on fourth-order ODE solver. We term our Time-Shift Sampler as *TS-*$\{\cdot\}$ with respect to the baseline sampling methods. For example, the time shift variant of DDIM is referred to as *TS*-DDIM. Following DDIM, we consider two types of time step selection procedures during sampling, namely *uniform* and *quadratic*. For $t_i < T$: (1) *uniform*: we select time steps such that $t_i = \lfloor ci \rfloor$, for a constant value $c$. (2) *quadratic*: we select time steps such that $t_i = \lfloor ci^2 \rfloor$, for a constant value $c$.

We report main results using pre-trained DDPM on CIFAR-10 (Krizhevsky, 2009) and CelebA 64×64 (Liu et al., 2015). Moreover, based on DDIM sampler, a comparison to ADM-IP (Ning et al., 2023) is made, which uses ADM (Dhariwal & Nichol, 2021) as the backbone model. More experiments can be found in the appendix. We conduct experiments for varying sampling time steps, namely 5, 10, 20, 50, and 100. We use the Frechet Inception Distance (FID) (Heusel et al., 2017) for evaluating the quality of the generated images. We further discuss the influence of window sizes and cutoff values in Sec. 5.4. More details can be found in Appendix C.

---

[2]The latter refers to the variance computed by considering corresponding elements across samples of $\hat{x}_{t-1}$.
[3]We discuss the influence of the window size and the cutoff value in Sec. 5.4.

# 5 RESULTS

## 5.1 MAIN RESULTS

| Dataset | Sampling Method | 5 steps | 10 steps | 20 steps | 50 steps | 100 steps |
|---|---|---|---|---|---|---|
| CIFAR-10 | DDIM (*quadratic*) | 41.57 | 13.70 | 6.91 | 4.71 | 4.23 |
| | *TS*-DDIM(*quadratic*) | **38.09 (+8.37%)** | **11.93 (+12.92%)** | **6.12 (+11.43%)** | **4.16 (+11.68%)** | **3.81 (+9.93%)** |
| | DDIM(*uniform*) | 44.60 | 18.71 | 11.05 | 7.09 | 5.66 |
| | *TS*-DDIM(*uniform*) | **35.13 (+21.23%)** | **12.21 (+34.74%)** | **8.03 (+27.33%)** | **5.56 (+21.58%)** | **4.56 (+19.43%)** |
| | DDPM (*uniform*) | 83.90 | 42.04 | 24.60 | 14.76 | 10.66 |
| | *TS*-DDPM (*uniform*) | **67.06 (+20.07%)** | **33.36 (+20.65%)** | **22.21 (+9.72%)** | **13.64 (+7.59%)** | **9.69 (+9.10%)** |
| | S-PNDM (*uniform*) | 22.53 | 9.49 | 5.37 | 3.74 | 3.71 |
| | *TS*-S-PNDM (*uniform*) | **18.81(+16.40%)** | **5.14 (+45.84%)** | **4.42 (+17.69%)** | **3.71 (+0.80%)** | **3.60 (+2.96%)** |
| | F-PNDM (*uniform*) | 31.30 | 6.99 | 4.34 | 3.71 | 4.03 |
| | *TS*-F-PNDM (*uniform*) | **31.11 (+4.07%)** | **3.88 (+44.49%)** | **3.60 (+17.05%)** | **3.56 (+4.04%)** | **3.86 (+4.22%)** |
| CelebA | DDIM (*quadratic*) | 27.28 | 10.93 | 6.54 | 5.20 | 4.96 |
| | *TS*-DDIM (*quadratic*) | **24.24 (+11.14%)** | **9.36 (+14.36%)** | **5.08 (+22.32%)** | **4.20 (+19.23%)** | **4.18 (+15.73%)** |
| | DDIM (*uniform*) | 24.69 | 17.18 | 13.56 | 9.12 | 6.60 |
| | *TS*-DDIM (*uniform*) | **21.32 (+13.65%)** | **10.61 (+38.24%)** | **7.01 (+48.30%)** | **5.29 (+42.00%)** | **6.50 (+1.52%)** |
| | DDPM (*uniform*) | 42.83 | 34.12 | 26.02 | 18.49 | 13.90 |
| | *TS*-DDPM (*uniform*) | **33.87 (+20.92%)** | **27.17 (+20.37%)** | **20.42 (+21.52%)** | **13.54 (+26.77%)** | **12.83 (+7.70%)** |
| | S-PNDM (*uniform*) | 38.67 | 11.36 | 7.51 | 5.24 | 4.74 |
| | *TS*-S-PNDM (*uniform*) | **29.77 (+23.02%)** | **10.50 (+7.57%)** | **7.34 (+2.26%)** | **5.03 (+4.01%)** | **4.40 (+7.17%)** |
| | F-PNDM (*uniform*) | 94.94 | 9.23 | 5.91 | 4.61 | 4.62 |
| | *TS*-F-PNDM (*uniform*) | **94.26 (+0.72%)** | **6.96 (+24.59%)** | **5.84 (+1.18%)** | **4.50 (+2.39%)** | **4.42 (+4.33%)** |

Table 1: Quality of the image generation measured with FID ↓ on CIFAR-10 (32×32) and CelebA (64×64) with varying time steps for different sampling algorithms.

In Table 1, we compare four sampling methods, namely DDIM, DDPM, S-PNDM and F-PNDM, and their Time-Shift variants on two datasets, where we vary the time steps and the time step selection procedures. We take larger window sizes for fewer sampling steps. The cutoff value is within the range of $[200, 400]$, and is dependent to the number of time steps we take. As expected, our Timer-Shift Sampler consistently improves the quality of the generated images with respect to that of the baseline sampling methods. We observe that for time steps less than 20 with *uniform* time step selection, the performance improvements are extremely significant as compared to the original sampling methods. To our surprise, even for very strong high-order sampling methods, our Time-Shift Sampler can still bring significant improvements. Notably, we obtain FID=3.88 with *TS*-F-PNDM on CIFAR-10 using 10 sampling steps, which is better than DDIM on CIFAR-10 for 100 sampling steps. Our method also manages to improve the performance of the baseline sampling methods for both *uniform* and *quadratic* time selection, showing the versatility of our method.

## 5.2 DISCUSSION ON EFFICIENCY AND PERFORMANCE

Our Time-Shift Sampler involves searching for suitable time steps using computed variance of the generated samples, which inevitably brings additional computation during sampling. To compare the efficiency of different sampling methods, we present the average sampling time[4] in Figure 5 by running each sampling method 300 times on CIFAR-10 using DDPM as backbone model. For all three sampling methods, the additional computation time during sampling with 5,10 and 20 sampling steps is negligible. The additional computation time is visually larger for 50 sampling steps. Yet, the actual additional computation time remains acceptable for a small backbone like DDPM. For example, *TS*-S-PNDM requires 9.71% more sampling time on average than S-PNDM for 5,10 and 20 steps. For 50 steps, *TS*-S-PNDM

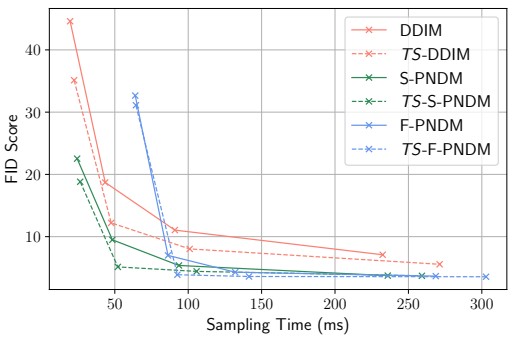

Figure 5: Sampling time VS FID on CIFAR-10 using DDPM as backbone with various sampling methods. We report the results of {5,10,20,50} sampling steps from left to right for each sampler, denoted with "×" symbol.

---

[4]Test conducted on an AMD EPYC7502 CPU and a RTX 3090 GPU using Pytorch time estimation API.

requires 9.86% more sampling time than S-PNDM. We report the detailed sampling time in Appendix F. The additional computation time is also dependant to the choice of the backbone, which we further elaborate in Sec. 5.3.

## 5.3 COMPARISON WITH TRAINING-REQUIRED METHOD ADM-IP

| Model | Sampling Method | 5 steps | 10 steps | 20 steps | 50 steps |
|---|---|---|---|---|---|
| ADM | DDIM | 28.98 | 12.11 | 7.14 | 4.45 |
| ADM-IP | DDIM | 50.58 (-74.53%) | 20.95 (-73.00%) | 7.01 (+1.82%) | **2.86 (+35.73%)** |
| ADM | *TS*-DDIM | **26.94 (+7.04%)** | **10.73 (+11.40%)** | **5.35 (+25.07%)** | 3.52 (+20.90%) |

Table 2: Performance comparison on CIFAR-10 with ADM and ADM-IP as backbone models.

Our method is also proven to be effective on alternative model architectures than DDPM. In this section we present the results on ADM (Dhariwal & Nichol, 2021), and the variant of ADM named ADM-IP (Ning et al., 2023), which tries to alleviate exposure bias in DPM by retraining ADM with added input perturbation. We conduct our Time-Shift Sampler using ADM as backbone model to show the merits of our method as compared to ADM-IP. We present the comparison on FID scores in Table 2 and the comparison on sampling time in Figure 6. The results indicate that when employing a small sampling step, which is favored in practice, ADM-IP performs much worse than the original ADM. As the number of sampling steps increases, the performance of ADM-IP improves. We manage

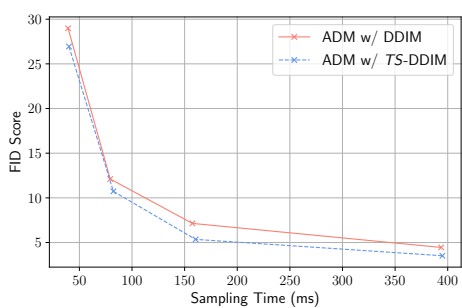

Figure 6: Sampling time VS FID on CIFAR-10 with ADM as backbone using DDIM and *TS*-DDIM for sampling.

to improve the performance of the ADM model with our method by a large margin. Note that we achieve these significant improvements without retraining the model as is required by ADM-IP. We also obtain roughly the same sampling time for DDIM and *TS*-DDIM with ADM as the backbone model. It makes our method more favorable in practice given merely zero additional cost for computation and significant performance improvements.

## 5.4 INFLUENCE OF WINDOW SIZES AND CUTOFF VALUES

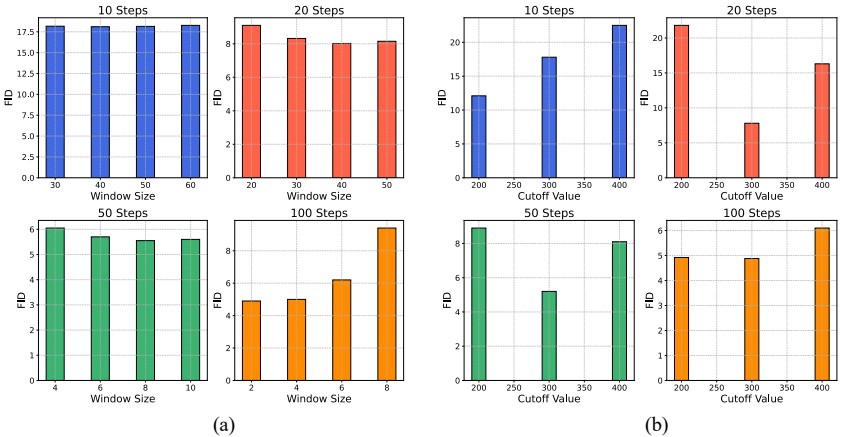

Figure 7: FID of generated CIFAR-10 images using *TS*-DDIM (*uniform*) with (a) various window sizes using cutoff value=300; (b) various cutoff values using window size= {40;30;8;2} for {10;20;50;100} steps.

We first conduct a study on the effect of window sizes. The results are presented in Figure 7 (a), where we fix the cutoff value=300, and vary the window sizes for different numbers of sampling steps. In the assessment of 10 and 20 sampling steps, larger window sizes are evaluated to ensure an

adequate search space for a suitable time step. And we adopt smaller window sizes for more sampling steps, given the limitation of step sizes. Figure 7 (a) illustrates that the Time-Shift algorithm is not sensitive to the selection of window size when using $10, 20$ and $50$ sampling steps. While when the number of sampling step is increased, such as $100$ steps, the algorithm become more sensitive to the window size, which could be attributed to the enhanced accuracy of per-step predictions achievable with smaller step sizes.

The influence of different cutoff values on the image generation performance on CIFAR-10 are presented in Figure 7 (b). From the density distribution plot of the sample variance as discussed in Sec. 2.2, we can have a good estimation of the range of the cutoff values to be within $[200, 400]$. While sampling with 10 steps, a smaller cutoff value (200) is preferred as compared to the scenarios with more sampling steps. One possible reason is that fewer sampling steps lead to larger step size, which makes more time-shift operations beneficial.

## 6 RELATED WORK

**Denoising Diffusion Probabilistic Model.** The denoising diffusion probabilistic model (DDPM) was first introduced by Sohl-Dickstein et al. (2015) and further advanced by Nichol & Dhariwal (2021), where they include variance learning in the model and optimize it with a new weighted variational bound. Song et al. (2020) further connect DDPMs with stochastic differential equations by considering DDPMs with infinitesimal timesteps. They also find that both score-based generative models (Song & Ermon, 2019) and that DDPMs can be formulated by stochastic differential equations with different discretization. Some variants of DDPMs are recently introduced, including the variational diffusion model (VDM) Kingma et al. (2021) that also learns the forward process, and consistency model (Song et al., 2023) and rectified flow model (Liu et al., 2022b), both of which aim to learn to generate natural images in one inference step.

DDPMs are initially applied in the pixel space achieving impressive success in generating images of high quality, however, they suffer from the significant drawbacks such as prolonged inference time and substantial training cost. Rombach et al. (2022b) propose to use a diffusion model in the latent image space, which drastically reduces the computational time and training cost. DDPMs have been widely used in different fields, for example controllable image generation (Rombach et al., 2022b; Choi et al., 2021; Zhang & Agrawala, 2023; Mou et al., 2023), language generation (Zhang et al., 2023b; Ye et al., 2023; Lin et al., 2022), image super-resolution (Saharia et al., 2022b) and video generation (Hu et al., 2023; Karras et al., 2023; He et al., 2022).

**DDPM Accelerator.** Though diffusion models have shown powerful generation performance, it normally takes hundreds of steps to generate high-quality images (Ho et al., 2020). To accelerate the generation speed, Song et al. (2021) propose the denoising diffusion implicit model (DDIM) which derives ordinary differential equations for the diffusion model showing the possibility of generating high-quality images in much less steps. Liu et al. (2022a) introduce numerical solvers to further reduce the number of generation steps. Bao et al. (2022b) find that the variance in the backward process can be analytically computed and improve the quality of the generated images. Bao et al. (2022a) further remove the diagonal variance matrix assumption and boost the image generation performance. Lu et al. (2022) leverage a high-order ordinary differential equations solver to reduce the generation steps of a diffusion model to 10, while maintaining the image quality. We follow this line of research and propose a method to better select time steps.

## 7 CONCLUSION

In this paper, we have proposed a novel method to alleviate the exposure bias problem in diffusion probabilistic models. Different from previous work, which tries to reduce exposure bias by retraining the diffusion model with extra noisy inputs, we demonstrate that this bias can be mitigated by identifying the timestep that most aligns with the subsequent predicted step. Furthermore, we theoretically derive the variance of this optimal next timestep. Leveraging these insights, we introduce a novel and training-free inference algorithm, the Time-Shifted Sampler, to alleviate exposure bias in diffusion models. Our algorithm can be seamlessly incorporated into existing sampling algorithms for diffusion models including DDPM, DDIM, S-PNDM and F-PNDM. Extensive experimental results prove the effectiveness of our method.

## 8 ACKNOWLEDGEMENTS

This work is funded by the CALCULUS project (European Research Council Advanced Grant H2020-ERC-2017-ADG 788506) the Flanders AI Research Program, and the China Scholarship Council.

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

## A  LIMITATION AND FUTURE WORK

In this work, we introduce the Time-Shift Sampler, a training-free mechanism devised to mitigate the exposure bias problem inherent to the diffusion model, thereby enhancing the quality of generated images. Despite its efficacy, our sampler suffers from the limitation that it introduces two parameters, i.e., window size and cutoff value. We estimate their values based on the statistical analysis of training data. However, more advanced methods to analytically derive the optimal value of these two parameters might be possible since both the cutoff value and the window size are related to the noise level of each step. We leave this further exploration to future work. We also foresee possibilities to use the concept of the Time-Shift Sampler in other Markovian processes where a reduction in processing steps is desired.

## B  FIGURE DETAILS

In this section, we present the detailed procedures to plot Figures 2, 3, and 4.

To plot Figure 2, we compute the variance at the instance level so that we can measure how much the variance of each sample varies as time progresses. Specifically, suppose we obtain a sample $x$ of size $3 \times 32 \times 32$, we first flatten it to 3072. Then we compute the variance of $x$ as $\frac{\sum_i^n (x_i - \bar{x})^2}{n-1}$, where $\bar{x}$ is the mean of $x$. Thus, for each sample we obtain a variance of its own, which leads to the density plot in Figure 2.

Figure 3 shows the mean square error (MSE) between the prediction and ground truth at each step in the backward process. Given an image denoted as $x_0$, by applying Equation 2 we could obtain a sequence of $x_t, t = 1, 2, \cdots, T - 1$. Taking each $x_t$ and the paired time step $t$ to run the backward process, we would obtain a sequence of predicted $\hat{x}_0^t, t = 1, 2, \cdots, T - 1$. Ideally, we would expect all these predicted $\hat{x}_0^t, t = 1, 2, \cdots, T - 1$, to be exactly equal to the ground truth $x_0$, as they are generated using the given $x_0$. However, this is not the case in practice. In our experiments, we found that only when $t < t_s$ (around 650 steps in our experiment using DDIM) we could obtain the original $x_0$ by running the backward process with paired $(x_t, t)$. For $t > t_s$, the image created using $(x_t, t)$ differs from the original $x_0$. This observation also reveals that the image generation process of diffusion models basically contains two stages. In the first stage, the model moves the Gaussian distribution towards the image distribution and no modes are presented at this stage, which means we can not know which images will be generated. In the second stage, the prediction shows that clear patterns and modes are presented. We can predict which images will be generated following the backward process. This observation led us to divide the error computation into two stages. The full explanation, including the equations, is shown in Figure 8

To plot Figure 4, we follow the method used in plotting Figure 3 to generate the ground truth example for each step and compute the MSE between predicted $\hat{x}_t$ and the ground truth $x_t$ and $x_{t_s}$.

## C  ADDITIONAL EXPERIMENTAL SETUP

Instead of generating many samples of $x_{t-1}$ to estimate $var(x_{t-1})$, which brings additional computational workload, we find that under some assumption (see derivation of Theorem 3.1 in Section J.1) the $var(x_{t-1})$ could be estimated using the inter variance of a single $x_{t-1}$, Thus during sampling, we compute the variance within each $x_{t-1}$.

Following DDIM (Song et al., 2021), we use two types of time step selection procedures, namely *uniform* and *quadratic*. For comparing different sampling methods, the architecture of our models follows the one of DDPM (Ho et al., 2020). For all datasets, we use the same pretrained models for evaluating the sampling methods. For CIFAR-10 and LSUN-bedroom, we obtain the checkpoints from the original DDPM implementation; for CelebA, we adopt the DDIM checkpoints. To carry out the comparison between ADM (Dhariwal & Nichol, 2021), ADM-IP (Ning et al., 2023) and our method, we choose the ADM architecture and the checkpoints from ADM-IP.

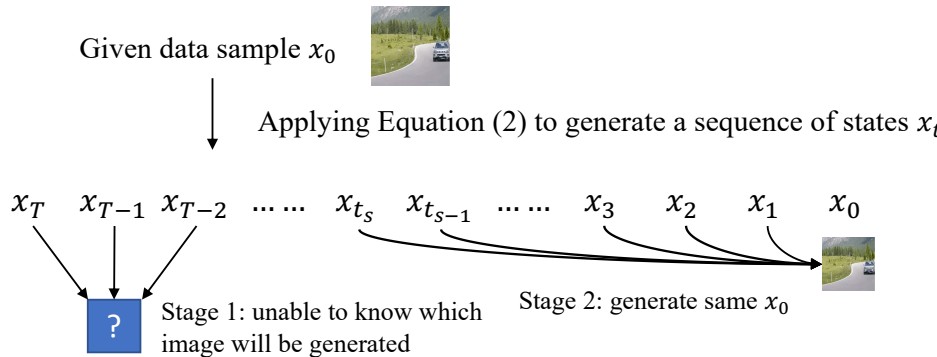

Given data sample $x_0$

Applying Equation (2) to generate a sequence of states $x_t$

$$x_T \quad x_{T-1} \quad x_{T-2} \quad ...\,... \quad x_{t_s} \quad x_{t_{s-1}} \quad ...\,... \quad x_3 \quad x_2 \quad x_1 \quad x_0$$

? Stage 1: unable to know which image will be generated

Stage 2: generate same $x_0$

1. The Computation of the MSE at Stage 1:

Applying Equation (2) to obtain the ground truth sequence: $x_{T-1}, ....., x_{t_s}$

Using $(x_T, T)$ as input to the model to generate a sequence of predicted $x_t$ till time $t_s$

$$Model(x_T, T) \rightarrow \hat{x}_{T-1}, \hat{x}_{T-2}, ......, \hat{x}_{t_s}$$

$$MSE_t = \frac{\sum_{i=1}^{n} ||\hat{x}_t - x_t||^2}{n}$$

2. The Computation of the MSE at Stage 2:

Applying Equation (2) to obtain $x_{t_s}$,

Using $(x_s, t_s)$ as input to the model to generate a sequence of predicted $\hat{x}_0^t$ at each time step.

$$MSE_t = \frac{\sum_{i=1}^{n} ||\hat{x}_0^t - x_0||^2}{n}$$

Figure 8: The detailed procedure for computing the MSE.

## D    ERROR ANALYSIS

We provide here more examples of prediction errors on training samples for different sampling steps for the different datasets. Figure 9 presents the prediction errors obtained in the CelebA dataset, which show similar patterns as those of the CIFAR-10 dataset as dipicted in Figure 3.

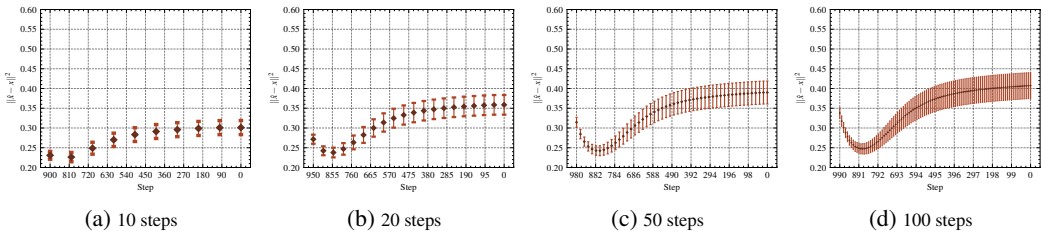

(a) 10 steps    (b) 20 steps    (c) 50 steps    (d) 100 steps

Figure 9: CelebA prediction errors of training samples for different numbers of sampling steps.

## E    TRAINING-INFERENCE DISCREPANCY

We provide more examples on the training-inference discrepancy for the different datasets in Figure 10, 19 and 20. For varying numbers of time steps, the same training-inference discrepancy pattern can be observed as in Figure 4. Specifically, for a certain backward step $t$, given a time step window $[t-w, t+w]$ surrounding $t$, there exists a time step $t_s$ that might couple better with the predicted next state $\hat{x}_{t-1}$. And as sampling progresses, the coupling effects for $\hat{x}_{t-1}$ become identical for surrounding time steps.

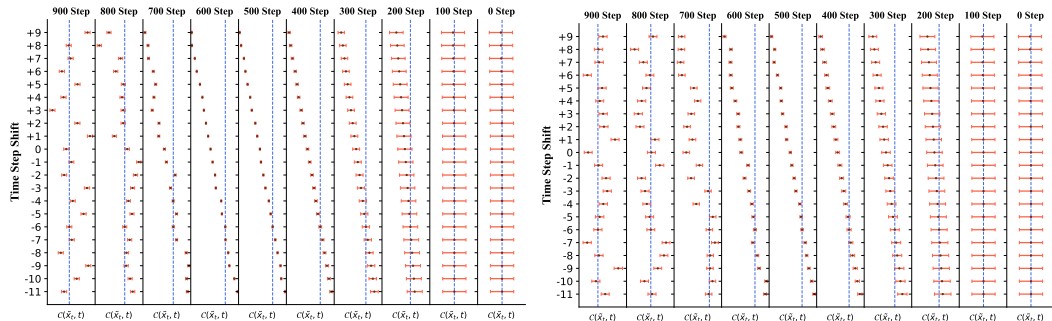

Figure 10: The training and inference discrepancy of DDIM with 10 sampling steps for window size=20: Left: CelebA dataset; Right: CIFAR-10 dataset.

## F SAMPLING TIME COMPARISON

We report a detailed comparison of sampling time using different sampling methods in Table 3 and 4.

| Sampling Method | 5 steps | 10 steps | 20 steps | 50 steps |
|---|---|---|---|---|
| DDIM | 19.57 ms | 43.46 ms | 90.90 ms | 232.47 ms |
| *TS*-DDIM | 22.30 ms (+13.9%) | 47.53 ms (+9.4%) | 101.01 ms (+11.1%) | 271.25 ms (+16.7%) |
| S-PNDM | 24.35 ms | 48.32 ms | 93.69 ms | 235.93 ms |
| *TS*-S-PNDM | 26.44 ms (+8.5%) | 52.08 ms (+7.78%) | 105.72 ms (+12.84%) | 259.19 ms (+9.86%) |
| F-PNDM | 63.93 ms | 86.23 ms | 132.03 ms | 268.65 ms |
| *TS*-F-PNDM | 64.39 ms (+0.7%) | 92.61 ms (+7.39%) | 141.38 ms (+7.08%) | 302.88 (+12.74%) |

Table 3: Sampling time comparison for different sampling methods on CIFAR-10 using DDPM as backbone.

| Sampling Method | 5 steps | 10 steps | 20 steps | 50 steps |
|---|---|---|---|---|
| ADM w/ DDIM | 39.26 ms | 79.30 ms | 157.43 ms | 393.77 ms |
| ADM w/ *TS*-DDIM | 40.09 ms (+2.10%) | 82.26 ms (+3.70%) | 160.45 ms (+1.92%) | 394.95 ms (+0.29%) |

Table 4: Sampling time comparison for DDIM and *TS*-DDIM on CIFAR-10 with ADM as backbone.

## G CASE STUDY

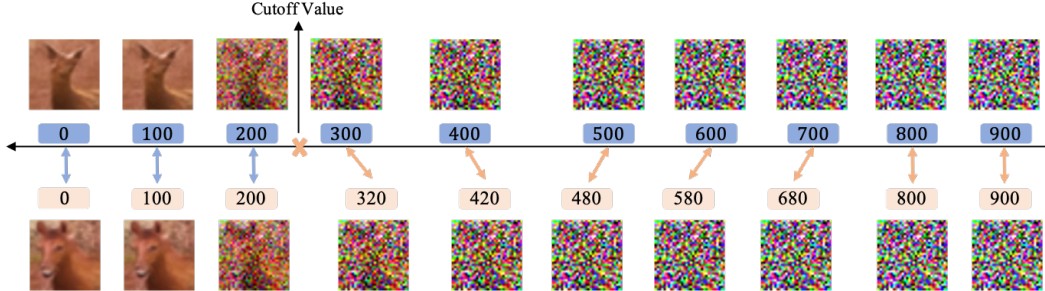

Figure 11: Example of generation process of *TS*-DDIM and DDIM on CIFAR-10. We use the horizontal black arrow to represent the time line, with the DDIM generation chain above it, *TS*-DDIM generation chain underneath it. We sample 10 steps with window $[t-20, t+20]$ and cutoff value=200.

The proposed novel sampling method is motivated by the theoretical and empirical analysis of the exposure bias problem in DDPM. Experimental results show that our sampling method can effectively improve the quality of the generated images compared to images generated with the original DDIM/DDPM models quantatively measured with the FID score. We present the comparison of the generation chain of *TS*-DDIM (our method) and DDIM in Figure 11. It can be seen that *TS*-DDIM shifted most of the time steps before reaching the cutoff value, which yields a much better generation quality of the image of a horse. Additional qualitative examples will be presented in the Appendix I.

We also visualize the selected timestep trajectory in Figure 12. Before the cutoff value 200, time shift happens to all the timesteps. This is especially visible for the time steps within the range of (300, 600), where more time shifts happen, and time steps can shift to both larger or smaller time steps within the window. Figure 12 demonstrates that most of the time shifting happens in the intermediate range of the backward process. This phenomenon might be due to the fact that at the initial stage (when $t$ is large), the samples are predominantly influenced by the initialized normal distribution, leading to minimal variance change. Conversely, in the later stage (when $t$ is small), the samples are predominantly composed of image data, containing most of the image information. This makes it easier for the model to make accurate predictions.

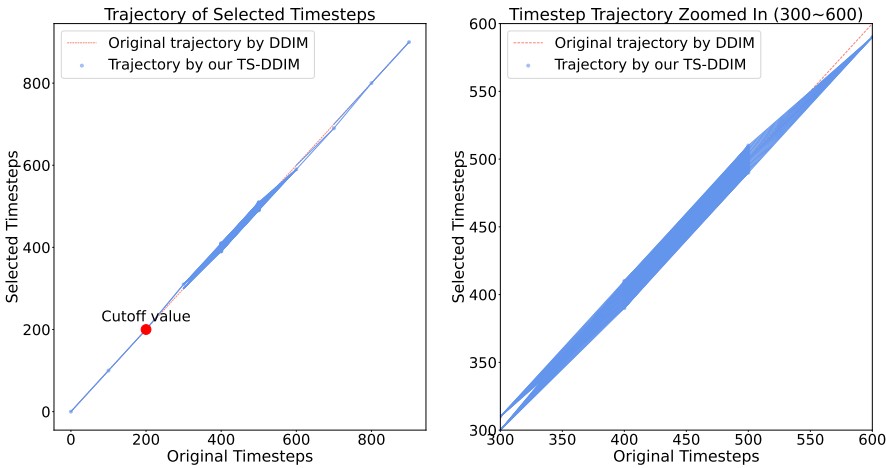

Figure 12: Comparison of timestep trajectory for *TS*-DDIM and DDIM on CIFAR-10 with DDPM as backbone using 10 sampling steps and uniform time selection. We use the red dashed line and blue line to present the time step trajectories for the original DDIM and our TS-DDIM, respectively. To improve visibility, we also zoom in for the time steps between 300 to 600.

## H  QUALITATIVE EXAMPLES

In this section we present the example images generated using different sampling methods with *uniform* time selection procedure for varying sampling time steps. Generated examples can be found in Figure 13, 14, 15, 16, 17 and 18. It can be seen that we can generate images with good quality for less than 10 sampling steps.

## I  ADDITIONAL RESULTS

We present more results obtained with the LSUN-bedroom dataset (Yu et al., 2016) in Table 5. Limited by computational resources, we do not properly tune the parameters, i.e., window size and cutoff values, for the LSUN-bedroom. We leave the exploration of a more efficient tuning strategy on high-resolution images for future work.

In Table 6 we compare our method with Analytic-DPM (Bao et al., 2022b). The results of Analytic-DPM are directly taken from the paper.

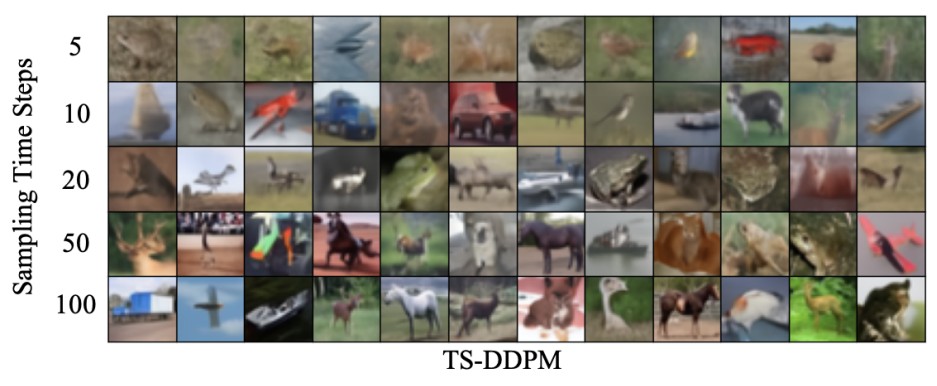

Figure 13: CIFAR-10 samples for varying time steps using TS-DDPM.

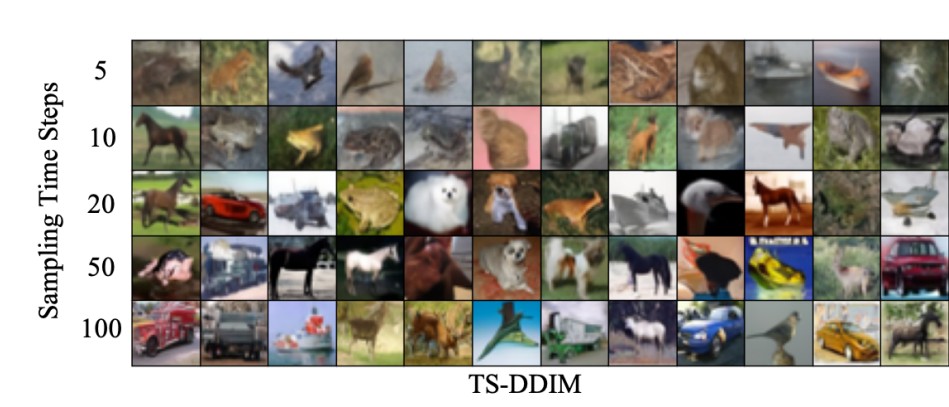

Figure 14: CIFAR-10 samples for varying time steps using TS-DDIM.

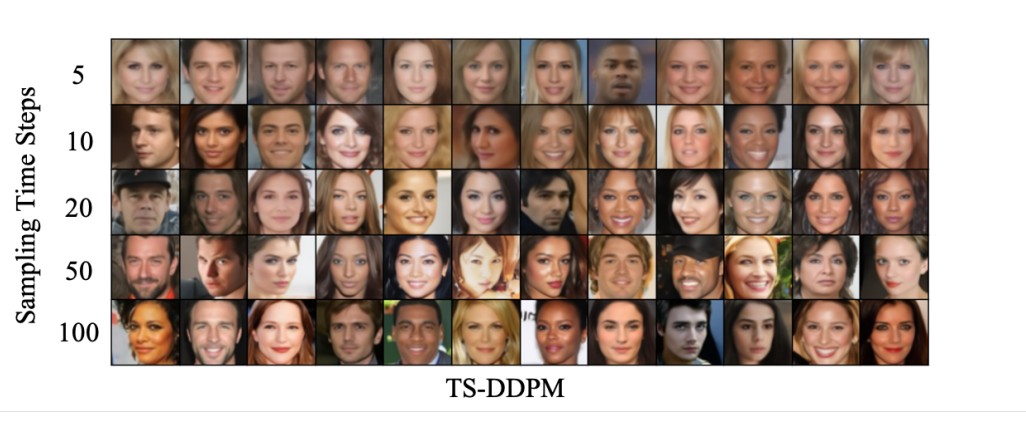

Figure 15: CelebA samples for varying time steps using TS-DDPM.

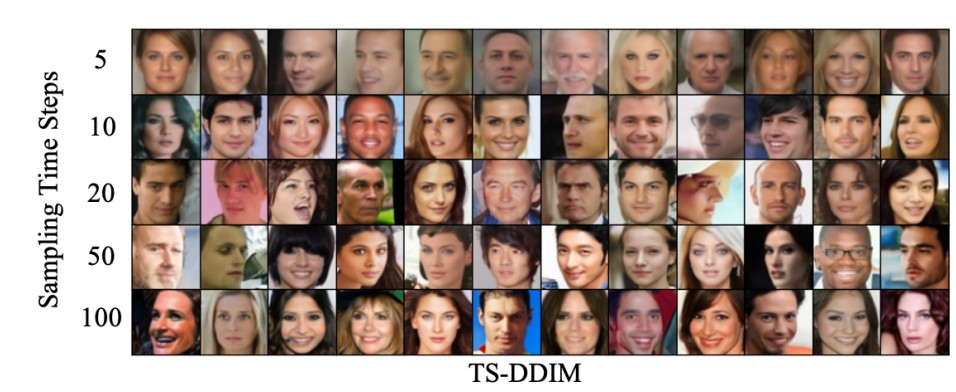

Figure 16: CelebA samples for varying time steps using TS-DDIM.

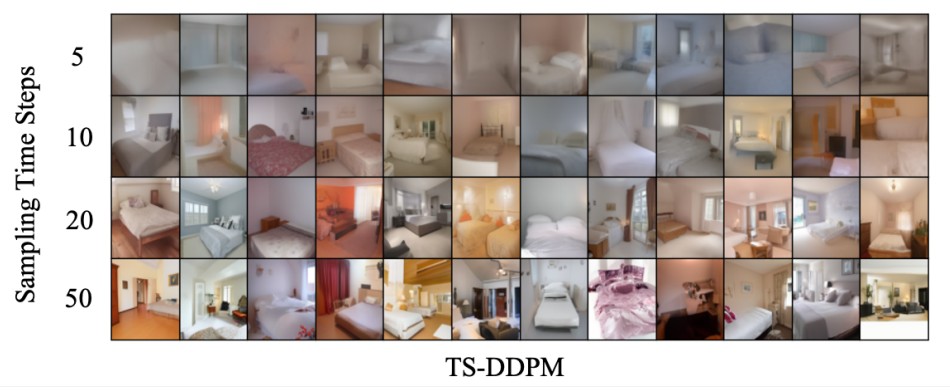

Figure 17: LSUN-bedroom samples for varying time steps using TS-DDPM.

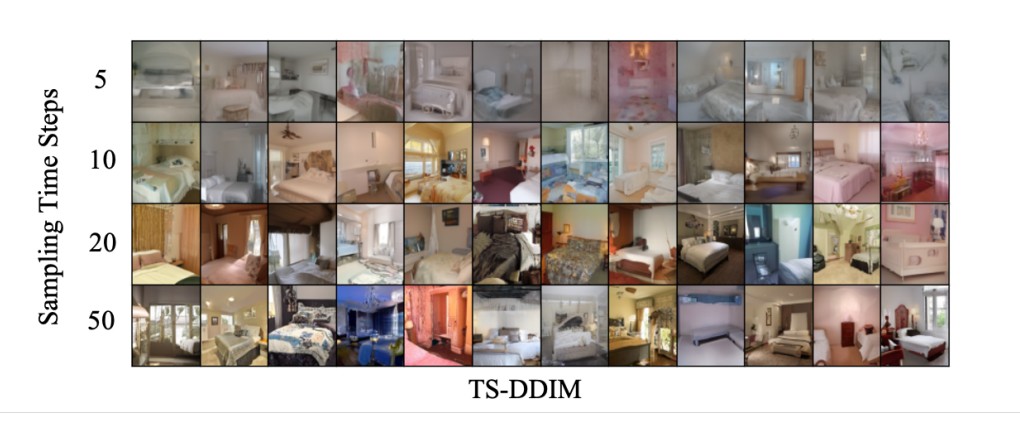

Figure 18: LSUN-bedroom samples for varying time steps using TS-DDIM.

| Dataset | Sampling Method | 5 steps | 10 steps | 20 steps | 50 steps | |
|---------|-----------------|---------|----------|----------|----------|---|
| | DDIM(*uniform*) | 52.29 | 16.90 | 8.78 | 6.74 | |
| LSUN-bedroom | *TS*-DDIM(*uniform*) | 51.57 | 16.66 | 8.29 | 6.90 | |
| | DDPM (*uniform*) | 85.60 | 42.82 | 22.66 | 10.79 | - |
| | *TS*-DDPM (*uniform*) | 79.05 | 32.47 | 15.40 | 10.20 | |

Table 5: Quality of the image generation measured with FID ↓ on LSUN-bedroom (256×256) with varying time steps for different sampling algorithms.

| Dataset | Sampling Method | 10 steps | 50 steps | 100 steps |
|---------|-----------------|----------|----------|-----------|
| CIFAR-10 | Analytic-DPM(DDIM) | 14.00 | 4.04 | 3.55 |
| | *TS*-DDIM | 11.93 | 4.16 | 3.81 |
| CelebA | Analytic-DPM(DDIM) | 15.62 | 6.13 | 4.29 |
| | *TS*-DDIM | 9.36 | 4.20 | 4.18 |

Table 6: Comparison of *TS*-DDIM with Analytic-DPM on CIFAR-10 (32×32) and CelebA (64×64)

Following (Xiao et al., 2022), we present the precision and recall results from our TS-DDIM and DDIM with ADM as backbone on CIFAR10. As shown in Table 7, our TS-DDIM tends to achieve much higher recall while maintaining the level of the precision obtained by DDIM.

| Sampling Method | 5 steps | | 10 steps | | 20 steps | | 50 steps | |
|-----------------|-----------|--------|-----------|--------|-----------|--------|-----------|--------|
| | Precision | Recall | Precision | Recall | Precision | Recall | Precision | Recall |
| ADM w/ DDIM | 0.59 | 0.47 | 0.62 | 0.52 | 0.64 | 0.57 | 0.66 | 0.60 |
| ADM w/ *TS*-DDIM | 0.57 | 0.46 | 0.62 | 0.55 | 0.64 | 0.60 | 0.65 | 0.62 |

Table 7: Comparison of precision and recall for DDIM and *TS*-DDIM on CIFAR-10 with ADM as backbone.

We also evaluate our method on the ImageNet dataset as presented in Table 8. We adopt ADM as the backbone model with classifier guidance.

| Sampling Method | 5 steps | 10 steps | 20 steps |
|-----------------|---------|----------|----------|
| DDIM | 67.63 | 13.74 | 6.83 |
| *TS*-DDIM | **39.47(+41.64%)** | **13.45(+2.11%)** | **6.57(+3.81%)** |

Table 8: Performance comparison with DDIM on ImageNet (64×64) with classifier guidance.

We further integrate our method into the DPM-solver (Lu et al., 2022) and DEIS (Zhang & Chen, 2023 ) to showcase its versatility. The results are presented in Table 9. Our model can still improve the performance of both the DPM-solver and DEIS samplers. However, in comparison to DDIM, our method again provides substantial improvements although the improvements for these samplers are a bit smaller than for the other tested samplers.This could be attributed to the fact that both the DPM-solver and DEIS utilize the particular structure of the semi-linear ODE, which already largely reduces the error of estimating $x_t$.

| Sampling Method | 5 steps | 10 steps | 20 steps |
|-----------------|---------|----------|----------|
| DPM-solver-2 | 32.30 | 10.92 | 4.30 |
| *TS*-DPM-solver-2 | **31.02(+3.96%)** | **9.82(+10.07%)** | **4.11(+4.42%)** |
| DEIS-order-2-*t*AB | 24.64 | 5.88 | 4.13 |
| *TS*-DEIS-order-2-*t*AB | **22.57(+8.40%)** | **5.41(+8.00%)** | **3.62(+12.30%)** |

Table 9: Performance comparison with DPM-solver and DEIS obtained on CIFAR-10 (32×32).

Finally, in Table 10, we report the performance of our method performed on text-to-image generation on MSCOCO val2017.

| Sampling Method | 10 steps | 20 steps |
|---|---|---|
| DDIM | 27.80 | 25.47 |
| *TS*-DDIM | **26.32 (+5.32%)** | **24.80 (+2.63%)** |

Table 10: Performance comparison with DDIM for text-to-image generation obtained on MSCOCO val2017.

## J DERIVATION

### J.1 DERIVATION OF THEOREM 3.1

In this section, we prove Theorem 3.1.

*proof.* We first find the optimal time step $t_s$ by minimizing the KL divergence between $\hat{x}_{t-1}$ and $x_{t_s}$. As $q(x_{t_s}|x_0) = \mathcal{N}(x_{t_s}|\sqrt{\overline{\alpha}_t}x_0, (1-\overline{\alpha}_t)\mathbf{I})$ and assume $p(\hat{x}_{t-1}|\hat{x}_t)$ is a probability density function of the distribution for $\hat{x}$ with mean $\hat{u}_{t-1}$ and covariance matrix $\hat{\Sigma}_{t-1}$, then according to Lemma 2. of Bao et al. (2022b), we have:

$$
\begin{aligned}
&\mathcal{D}_{KL}(p(\hat{x}_{t-1}|\hat{x}_t)||q(x_{t_s}|x_0)) \\
&= \mathcal{D}_{KL}(\mathcal{N}(x|\hat{\mu}_{t-1}, \hat{\Sigma}_{t-1})||\mathcal{N}(\mu_{t_s}, \Sigma_{t_s})) + \mathcal{H}(\mathcal{N}(x|\hat{\mu}_{t-1}, \hat{\Sigma}_{t-1})) - \mathcal{H}(p(\hat{x}_{t-1}|\hat{x}_t)) \\
&= \frac{1}{2}(\log(|\Sigma_{t_s}|) + Tr(\Sigma_{t_s}^{-1}\hat{\Sigma}_{t-1}) + (\hat{\mu}_{t-1} - \mu_{t_s})\Sigma_{t_s}^{-1}(\hat{\mu}_{t-1} - \mu_{t_s})^T) + C \\
&= \frac{1}{2}(d\log(1-\overline{\alpha}_{t_s}) + \frac{d}{1-\overline{\alpha}_{t_s}}Tr(\hat{\Sigma}_{t-1}) + \frac{1}{1-\overline{\alpha}_{t_s}}||\hat{\mu}_{t-1} - \mu_{t_s}||^2) + C
\end{aligned}
\tag{9}
$$

where $C = \frac{1}{2}d + \mathcal{H}(\mathcal{N}(x|\hat{\mu}_{t-1}\hat{\Sigma}_{t-1})) - \mathcal{H}(p(\hat{x}_{t-1}|\hat{x}_t)) - \frac{1}{2}\log(\frac{1}{|\hat{\Sigma}_{t-1}|})$ and $d$ is the dimension of $x_0$. Denoted $\mu_{t-1}$ as the ground truth of the mean of the distribution of $q(x_{t-1})$ and according to Equation 2, we have $\mu_{t-1} = \sqrt{\overline{\alpha}_{t-1}}x_0$. $\hat{\mu}_{t-1}$ can be rewritten as :

$$
\hat{\mu} = \mu_{t-1} + e = \sqrt{\alpha_{t-1}}x_0 + e
\tag{10}
$$

Here, $e$ is the network prediction error. Since $\mu_{t_s} = \sqrt{\overline{\alpha}_{t_s}}x_0$, Equation 9 can be rewritten as:

$$
\begin{aligned}
&\mathcal{D}_{KL}(p(\hat{x}_{t-1}|\hat{x}_t)||q(x_{t_s}|x_0)) \\
&= \frac{1}{2}(d\log(1-\overline{\alpha}_{t_s}) + \frac{d}{1-\overline{\alpha}_{t_s}}Tr(\hat{\Sigma}_{t-1}) + \frac{1}{1-\overline{\alpha}_{t_s}}||(\sqrt{\overline{\alpha}_{t-1}} - \sqrt{\overline{\alpha}_{t_s}})x_0 + e||^2) + C
\end{aligned}
\tag{11}
$$

if $t_s$ is close to $t-1$, then $\sqrt{\overline{\alpha}_{t-1}} - \sqrt{\overline{\alpha}_{t_s}} \approx 0$. We have

$$
\mathcal{D}_{KL}(p(\hat{x}_{t-1}|\hat{x}_t)||q(x_{t_s}|x_0)) \approx \frac{1}{2}(d\log(1-\overline{\alpha}_{t_s}) + \frac{d}{1-\overline{\alpha}_{t_s}}Tr(\hat{\Sigma}_{t-1}) + \frac{1}{1-\overline{\alpha}_{t_s}}||e||^2) + C
\tag{12}
$$

We further calculate the derivative of $D_{KL}$ with respect to $\Sigma_{t_s} = 1 - \overline{\alpha}_{t_s}$. We know that $D_{KL}$ gets its minimum at

$$
\Sigma_{t_s} = Tr(\hat{\Sigma}_{t-1}) + \frac{1}{d}||e||^2
\tag{13}
$$

We next estimate the $\hat{\Sigma}_{t-1}$ in Equation 13. Assuming each pixel of image $P \in R^{w \times h}$ follows distribution $\mathcal{N}(\mu_i, \sigma)$ with $\sigma$ being the variance, and $p_i \perp p_j$ if $i \neq j$, then the covariance of $P$ is $\sigma\mathbf{I}$ and we have:

$$
\begin{aligned}
\sigma_{t-1} &= \frac{(\sum_i(p_i - \overline{p})^2)}{d-1} \\
&= \frac{\sum_i(p_i^2 + \overline{p}^2 - 2p_i\overline{p})}{d-1} \\
&= \frac{\sum_i(p_i^2) - d\overline{p}^2}{d-1}
\end{aligned}
\tag{14}
$$

Taking expectation on both sides, we achieve

$$
\begin{aligned}
\mathbb{E}[\sigma_{t-1}] &= \frac{\sum_i(\mathbb{E}[p_i^2]) - d\mathbb{E}[\bar{p}^2]}{d-1} \\
&= \frac{\sum_i(\sigma + \mu_i^2)}{d-1} - \frac{d}{d-1}\mathbb{E}[(\frac{\sum_i p_i}{d})^2] \\
&= \frac{d\sigma}{d-1} + \frac{\sum_i \mu_i^2}{d-1} - \frac{d}{d-1}\mathbb{E}[(\frac{\sum_i p_i}{d})^2]
\end{aligned}
\tag{15}
$$

The last term on the RHS of Equation 15 can be rewritten as

$$
\begin{aligned}
\frac{d}{d-1}\mathbb{E}[(\frac{\sum_i p_i}{d})^2] &= \frac{d}{d-1}\frac{1}{d^2}(\sum_i \mathbb{E}(p_i)^2 + \sum_{i \neq j}\mathbb{E}(p_i)\mathbb{E}(p_j)) \\
&= \frac{d}{d-1}\frac{1}{d^2}(\sum_i((\mathbb{E}(p_i))^2 + \sigma) + \sum_{i \neq j}\mathbb{E}(p_i)\mathbb{E}(p_j)) \\
&= \frac{\sigma}{d-1} + \frac{1}{d(d-1)}(\sum_i(\mathbb{E}(p_i))^2 + \sum_{i \neq j}\mathbb{E}(p_i)\mathbb{E}(p_j)) \\
&= \frac{\sigma}{d-1} + \frac{1}{d(d-1)}(\sum_i \mu_i)^2
\end{aligned}
\tag{16}
$$

By combining Equation 15 and Equation 16 and denoting $\bar{\mu} = \frac{\sum_i \mu_i}{d}$ we have

$$
\begin{aligned}
\mathbb{E}[\sigma_{t-1}] &= \frac{d\sigma}{d-1} + \frac{\sum_i \mu_i^2}{d-1} - \frac{\sigma}{d-1} - \frac{(\sum_i \mu_i)^2}{d(d-1)} \\
&= \sigma + \frac{\sum_i \mu_i^2}{d-1} - \frac{(\sum_i \mu_i)^2}{d(d-1)} \\
&= \sigma + \frac{\sum_i(\mu_i^2) - d\bar{\mu}^2}{d-1} \\
&= \sigma + \frac{\sum_i(\mu_i^2 - 2\mu_i\bar{\mu} + \bar{\mu}^2)}{d-1} \\
&= \sigma + \frac{\sum_i(\mu_i - \bar{\mu})^2}{d-1}
\end{aligned}
\tag{17}
$$

Here $\bar{\mu}$ is the mean of $\mu_i$ and $\mu_i$ is the mean of the distribution of each pixel in $\hat{x}_{t-1}$ at time step $t-1$. The ground truth $x_{t-1} \sim \mathcal{N}(\sqrt{\bar{\alpha}_{t-1}}x_0, (1 - \bar{\alpha}_{t-1})\mathbf{I})$, thus $u^{gt} = \sqrt{\bar{\alpha}_{t-1}}x_0$. In practice, the $x_0$ is normalized to stay in the range of $-1$ to $1$, and $\sqrt{\bar{\alpha}_t}$ is close to zero when $t$ is large. Define $\zeta$ as the difference between $\mu$ and $\bar{\mu}$ and denote that $\zeta_i^{gt} = u_i^{gt} - \bar{\mu}^{gt}$ and $\hat{\zeta}^i = \mu_i - \bar{\mu}$, then when $t$ is large we have $\zeta_i^{gt} \approx 0$. Considering the network prediction error, we reach

$$
\hat{\zeta}^i = \zeta_i^{gt} + e_i \approx e_i
\tag{18}
$$

Thus Equation 17 can be rewritten as

$$
\mathbb{E}[\sigma_{t-1}] = \sigma + \frac{\sum_i(e_i)^2}{d-1} = \sigma + \frac{||e||^2}{d-1}
\tag{19}
$$

Multiplying $\mathbb{I}$ on both sides and taking trace

$$
d\sigma_{t-1} \approx d\sigma + \frac{d||e||^2}{d-1}
\tag{20}
$$

Here we assume the sample variance is approximately equal to its expected value because the dimension of the image is usually large. Bring Equation 20 to Equation 13

$$
\sigma_{t_s} = \sigma_{t-1} - \frac{||e||^2}{d(d-1)}
\tag{21}
$$

In the above derivation, we assume that when $t$ is large Equation 21 holds. This assumption corresponds to the cutoff mechanism in our proposed algorithm, where we stop conducting time shift when $t$ is small, as the assumption does not hold and we are not able to estimate the $\hat{\Sigma}_{\hat{x}_{t-1}}$ in Equation 13.

## J.2    ANALYTICAL ESTIMATION OF WINDOW SIZE

In this section, we derive the bounds of window size $w$ with optimal time step $t_s \in [t-1-w/2, t-1+w/2]$. In Algorithm 3, we predefine a window size and search the optimal time step $t_s$ around time step $t-1$ within $w$. In the above derivation in Section J.1, after Equation 11, we assume $t_s$ is close to $t-1$, thus we can omit the term $\sqrt{\bar{\alpha}_{t-1}} - \sqrt{\bar{\alpha}_{t_s}}$, and the following derivations (Equations 12 to 21) give us the estimated variance of the optimal time step $t_s$. In order to estimate the window size $w$, we first relax our assumption. Instead of directly assuming $t_s$ is close to $t-1$, in the last term of Equation 11 we assume that the norm of $(\sqrt{\bar{\alpha}_{t-1}} - \sqrt{\bar{\alpha}_{t_s}})x_0$ is sufficiently smaller than the norm of $\gamma e$, where $0 < \gamma \ll 1$. Thus we have:

$$||\sqrt{\bar{\alpha}_{t-1}} - \sqrt{\bar{\alpha}_{t_s}}|| \, ||x_0|| \leq \gamma ||e||$$
$$(||\sqrt{\bar{\alpha}_{t-1}} - \sqrt{\bar{\alpha}_{t_s}}||)^2 \leq \gamma^2 \frac{||e||^2}{||x_0||^2} \tag{22}$$

$$\bar{\alpha}_{t-1} + \bar{\alpha}_{t_s} - 2\sqrt{\bar{\alpha}_{t-1}\bar{\alpha}_{t_s}} - \gamma^2 \frac{||e||^2}{||x_0||^2} \leq 0 \tag{23}$$

Solving Equation 23, we obtain:

$$\sqrt{\bar{\alpha}_{t_s}} \geq \frac{2\sqrt{\bar{\alpha}_{t-1}} - \sqrt{4\bar{\alpha}_{t-1} - 4(\bar{\alpha}_{t-1} - \gamma^2 \frac{||e||^2}{||x_0||^2})}}{2}$$
$$\sqrt{\bar{\alpha}_{t_s}} \leq \frac{2\sqrt{\bar{\alpha}_{t-1}} + \sqrt{4\bar{\alpha}_{t-1} - 4(\bar{\alpha}_{t-1} - \gamma^2 \frac{||e||^2}{||x_0||^2})}}{2} \tag{24}$$

In Equation 24, $e$ is the network prediction error and $\gamma$ is a predefined value. $e$ could be estimated using a small amount of data or a trained error prediction network similar to the work of Bao et al. (2022a). $\bar{\alpha}_t$ on the LHS is the predefined noise schedule. Since $\bar{\alpha}_t$ is a monotonic function with respect to $t$, denoted as $\bar{\alpha}_t = f(t)$, for a given $\bar{\alpha}_t$, the corresponding time step $t$ could be obtained through the inverse function of $f$, that is $t = f^{-1}(\bar{\alpha}_t)$. Thus, the bounds of window size $w$ could be estimated from the bounds of $\bar{\alpha}_t$ through Equation 24. To obtain the bounds of $w$, one could first compute the right hand sides of Equation 24 for the predefined noise schedule of a given diffusion model. Then for time step $t$, one could find the largest and smallest $t_s$, denoted as $t_s^{max}$ and $t_s^{min}$ respectively, satisfying Equation 24. The $w$ is then bounded by the $2 \times min(t_s^{max}, t_s^{min})$. If the above condition holds, then the last term of Equation 11 is dominated by $e$, and $(\sqrt{\bar{\alpha}_{t-1}} - \sqrt{\bar{\alpha}_{t_s}})x_0$ can be ignored in above derivations (Equations 12 to 21).

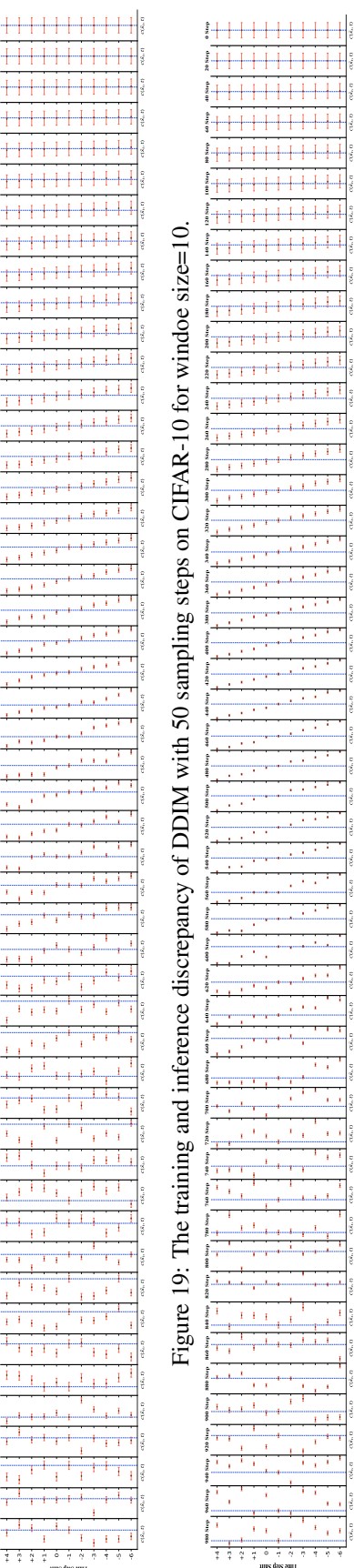

Figure 19: The training and inference discrepancy of DDIM with 50 sampling steps on CIFAR-10 for windoe size=10.

Figure 20: The training and inference discrepancy of DDIM with 50 sampling steps on CelebA for windoe size=10.

