# OpenReview forum: "Alleviating Exposure Bias in Diffusion Models through Sampling with Shifted Time Steps"
_ICLR.cc/2024/Conference — ICLR 2024 poster_

### Official Review · Reviewer_zG28 · 2023-10-28

**Soundness:** 3 good
**Presentation:** 3 good
**Contribution:** 2 fair
**Rating:** 6
**Confidence:** 4

**Summary:**

To address the issue of exposure bias, this paper proposes a sampling method called the Time-Shift Sampler. Specifically, this method is based on the observation that within a window of size w around the time step t, there exists a better time step ts that corresponds to a closer match between the variance of the true samples and the predicted samples xt. Moreover, the feasibility of this method is theoretically validated by the authors. To validate the effectiveness of the proposed method, extensive experiments are conducted on multiple models and datasets, and the results strongly demonstrate its efficacy. Importantly, compared to other relevant papers addressing exposure bias, this article offers the advantage of not requiring retraining and incurring minimal additional costs.

**Strengths:**

1. This paper investigate an important problem which ignored by previous works, dubbed exposure bias, and propose an interesting method to remedy it.
2. The analysis for the exposure bias problem helps to understand the proposed method, Time-Shift Sampler.
3. Time-Shift Sampler does not require fine-tune the pre-trained diffusion models and incurs minimal additional costs, while effectively mitigating exposure bias.
4. Moreover, Time-Shift Sampler enables to combine with various diffusion model, which demonstrates a good scalability.

**Weaknesses:**

1. Due to the conditions imposed by the theoretical derivation, amost 10% sampling time can not be simply ignored.
2. As mentioned 'seamlessly integrated to existing sampling algorithms', how about the performance combined with the DPM-solver [1], DEIS [2].
3. In my humble opinion, the analysis of the exposure bias problem is empirically not theoretically as mentioned in the 'contribution'.
4. There is no obvious advantage over the recent training-free sampling method.

[1] C. Lu, Y. Zhou, F. Bao, J. Chen, C. Li, and J. Zhu. DPM-solver: A fast ode solver for diffusion probabilistic model sampling in around 10 steps. Advances in Neural Information Processing Systems, 35:5775–5787, 2022.
[2]  Q. Zhang and Y. Chen. Fast sampling of diffusion models with exponential integrator. International Conference on Learning Representations, 2023.

**Questions:**

1. Can this method combine with other training-free sampling methods, such as DPM-sovler, DEIS?
2. How is the performance on ImageNet?
3. Is there exits a analytical solution for the window size?

---

> ### Author Response · Authors · 2023-11-18
> **Response to Reviewer zG28 (part1)**
>
> Thank you for your insightful comments and questions. Please find our response below (part1):
>
> > **[W1] Due to the conditions imposed by the theoretical derivation, almost 10% sampling time can not be simply ignored.**
>
> Regarding the additional +9.5% sampling time when sampling with DDPM on CIFAR-10:
>
> 1. In practice, we believe it is acceptable, for example, when we are sampling with 20 steps, it only requires less than 10ms additional sampling time.
>
> 2. We would not consider the conditions imposed by the theoretical derivation as the cost.
> In fact, our method requires nearly no additional sampling time when working with a bigger backbone ADM[1] as shown in Figure 6 and Table 2.  It makes our method stand out in nearly a “free lunch” situation, where we make significant improvements in performance without sacrificing efficiency.
> The additional computation we introduce to the sampling method only involves (1) computation of the sample variance (2) calculating $\mathop{\arg\min}(\cdot)$ within a small window size of timesteps. These two computations can be quickly achieved in modern GPUs.
>
> ***
>
> > **[W3] In my humble opinion, the analysis of the exposure bias problem is empirically not theoretically as mentioned in the 'contribution'.**
>
> Thank you for acknowledging our sound empirical findings on the exposure bias problem. Regarding the theoretical aspect of the problem, we would like to point out that:
>
> 1. We derive the analytic form of the optimal variance $\theta_s$ for the optimal time step $t_s$ corresponding to any $t-1$ in Theorem 3.1.
>
> 2. We show that the cutoff mechanism is a necessity to reach the derivation of Equation 21 (in Appendix J). It provides the theoretical foundation for the cutoff values.
>
> 3. The purpose of the selection window of timesteps is already well embedded in our theorem.
>
> 4. We further add the derivation of the analytic estimation for window size in Appendix J.2 as you suggested.
> We believe our theoretical contribution to the problem is just as important as our empirical findings.
>
> ***
>
> > **[W4] There is no obvious advantage over the recent training-free sampling method.**
>
> Limited to resources, we cannot test all the existing sampling methods. In Table 1, we already show the significant improvements obtained when applying our method combined with S-PNDM and F-PNDM [1], arguably two of the most performant sampling methods at this moment. Also as requested, we conduct experiments with  DPM-solver [2] and DEIS [3] as shown below (Q1). These experiments show that our method could be  seamlessly integrated to existing sampling algorithms. Thus, it is suitable to see our method as a way to further augment other sampling methods.
>
> Furthermore, our contributions go beyond just a sampling method. We show that the exposure bias problem of diffusion models could be alleviated without retraining the model, which could inspire future research on designing new samplers. We consider the theoretical and empirical findings regarding the exposure bias problem that we present in the paper  to be pioneering to the field.
>
>
> [1] Liu et al. Pseudo numerical methods for diffusion models on manifolds. ICLR22
>
> [2] Lu et al. DPM-solver: A fast ode solver for diffusion probabilistic model sampling in around 10 steps. NeurIPS22
>
> [3] Zhang and Chen. Fast sampling of diffusion models with exponential integrator. ICLR23

---

> ### Author Response · Authors · 2023-11-18
> **Response to Reviewer zG28 (part2)**
>
> Please find part2 of our response below:
>
> > **[W2, Q1] Can this method combine with other training-free sampling methods, such as DPM-solver, DEIS?**
>
> We report the results of our method combined with DPM-solver-2 and DEIS-order-2 on CIFAR-10 in the table below. Limited to resources and rebuttal time, we report the results using 5, 10 and 20 sampling steps. (The results can also be found in Table 9 in Appendix I of the revised manuscript.) Our model can still improve the performance of both DPM-solver and DEIS samplers. However, in comparison to DDIM, our method again provides substantial improvements although the improvements for these samplers are a bit smaller than for the other tested samplers.This could be attributed to the fact that both the DPM-solver and DEIS utilize the particular structure of the semi-linear ODE, which already largely reduces the error of estimating $x_t$.
>
>  | Sampling Method | 5 steps | 10 steps | 20 steps|
> |---|---|---|---|
> |DPM-solver-2 | 32.30 | 10.92 | 4.30 |
> |*TS*-DPM-solver-2| **31.02 (+3.96%)** | **9.82 (+10.07%)** | **4.11 (+4.42%)** |
> |DEIS-order-2-*t*AB | 24.64 | 5.88 | 4.13 |
> |*TS*-DEIS-order-2-*t*AB | **22.57 (+8.40%)** | **5.41 (+8.00%)** | **3.62 (+12.30%)** |
>
> ***
>
> > **[Q2] How is the performance on ImageNet?**
>
> We adopt ADM backbone trained on ImageNet 64$\times$64 with classifier guidance [1]. Due to limited resources and rebuttal time, we present results for DDIM and our *TS*-DDIM using 5, 10 and 20 steps. Once more, our method provides substantial improvements. The results are presented in the table below. (The results can also be found in Table 8 in Appendix I of the revised manuscript.)
>
> | Sampling Method | 5 steps | 10 steps | 20 steps|
> |---|---|---|---|
> |DDIM | 67.63 | 13.74 | 6.83 |
> |*TS*-DDIM| **39.47(+41.64%)** | **13.45(+2.11%)** | **6.57(+3.81%)** |
>
> [1] Dhariwal and Nichol. Diffusion Models Beats GAN on Image Synthesis. NeurIPS21
>
> ***
>
> > **[Q3] Is there exits a analytical solution for the window size?**
>
> Thank you for the question. We provide the derivation for the analytical estimation of the window size in Appendix J.2.

---

> > ### Comment · Reviewer_zG28 · 2023-11-20
> >
> > Your brilliant rebuttal is greatly appreciated! The responses are quite convincing. As a result, I have decided to improve the score.

---

> ### Author Response · Authors · 2023-11-21
>
> Dear Reviewer zG28,
>
> Thank you very much for reconsidering our paper and for increasing your score. We deeply appreciate your thoughtful feedback and the time you invested in our work.
>
> Best regards,
>
> The authors

---

### Official Review · Reviewer_rQd1 · 2023-10-31

**Soundness:** 3 good
**Presentation:** 3 good
**Contribution:** 3 good
**Rating:** 6
**Confidence:** 4

**Summary:**

This paper examines the exposure bias of diffusion models and proposes a straightforward and efficient method to address it. The study presents clear experiments and motivation to elucidate both the exposure bias phenomenon and its solution.

**Strengths:**

1. This paper analyzes the exposure bias of diffusion models, presenting clear visual results and detailed analysis.
2. This paper presents a simple, effective, and training-free solution for exposure bias.
3. The proposed solution can be applied to both DDPM-like and DDIM-like methods, providing potential benefits for future acceleration work.

**Weaknesses:**

The selection of window sizes and cutoff values is primarily based on limited experience.

**Questions:**

How can we design a more effective strategy for selecting window sizes and cutoff values when dealing with random datasets and image sizes?

---

> ### Author Response · Authors · 2023-11-18
> **Response to Reviewer rQd1**
>
> Thank you for your insightful comments and questions. Please find our response below:
>
> > **[W1] The selection of window sizes and cutoff values is primarily based on limited experience.**
>
> The characteristics of our empirical findings indicate that the selection procedure of window sizes and cutoff values are  robust and can be extended to different settings.
> The empirical findings are (1) very intuitive. For example, more sampling steps require smaller window sizes. It is well motivated by the fact the violation of Gaussian assumption is less a problem for more sampling steps. And the cutoff value is closely related to the changes of sample variance along different timesteps as presented in Figure 2. (2) The empirical findings are showing very similar patterns across datasets, models and sampling methods. As also reflected by the results obtained in various settings, given the same number of sampling steps, we find that the optimal values for window sizes and cutoff values always fall into the same range. For a more analytic solution of window size, please refer to the answer to the Q3 of reviewer zG28 or the Appendix J.2.
>
>
> ***
>
> > **[Q1] How can we design a more effective strategy for selecting window sizes and cutoff values when dealing with random datasets and image sizes?**
>
> It is intriguing to study the design of a more systematic and effective selection strategy for window sizes and cutoff values. As stated before, we find the optimal values for window sizes and cutoff values show remarkably similar patterns across datasets and models.
> Specifically, as stated in Section 5.4, (1) one should opt for large window sizes (e.g. 40, 80) for fewer steps(e.g. 5,10 steps) and very small window sizes (e.g. 2) for more steps (e.g.50,100 steps). This holds true for all our experiments. (2) The optimal cutoff values are always within the range of [200,400]. Under different experimental settings, only less than 5% of the cases in our experiments the optimal cutoff value falls outside the range of [200, 400]. For a more analytic solution of window size, please refer to the answer to the Q3 of reviewer zG28 or the Appendix J.2.

---

### Official Review · Reviewer_YoEn · 2023-10-31

**Soundness:** 2 fair
**Presentation:** 2 fair
**Contribution:** 2 fair
**Rating:** 6
**Confidence:** 4

**Summary:**

This work studies the problem of exposure bias in diffusion models and proposes a training-free method to mitigate exposure bias during sampling. The paper first reasons why mitigating exposure bias in diffusion models could result in improved sampling by empirically demonstrating the accumulation of error at different time steps. To this end, a quantity $C(x_t, t)$, called input couple for trained DPM, is used to capture the discrepancy between ground truth and network predictions. The core idea is that there might be an alternate time step $t_s$ that might align better with the next state $\hat{x}_{t-1}$  predicted by the network. This assumption is empirically demonstrated for different datasets and for different choices of time steps.

In order to find this alternate optimal time step $t_s$, this work derives the variance of this optimal tilmestep. This can then be used to empirically determine $t_s$ on-the-fly during sampling. This step can be seamlessly integrated into existing sampling methods like DDIM, DDPM, and PNDM. Further, this does not require any fine-tuning or training of diffusion models. However, this comes at the cost of some minor additional overhead in terms of sampling time. Overall, this method gives consistent improvements over baselines of DDIM, DDPM and PNDM in terms of FID scores.

**Strengths:**

1. The proposed method for alleviating exposure bias is training-free unlike previously proposed methods.
2. The proposed method results in consistent improved performance in terms of FID score compared to baselines of DDIM, DDPM, and PNDM sampling (See Table 1), as well as prior works like ADM-IP.
3. The primary contribution of this work which is empirical demonstration of the fact that correction of exposure bias can be done without retraining diffusion models is valuable.

**Weaknesses:**

1. The proposed method is training-free but introduces minor overhead in sampling time of diffusion models. Further, as the number of sampling steps increases, the method seems to be sensitive to the choice of hyperparameters like window size. At smaller number of sampling steps, the method is also a bit sensitive to cutoff time. (See Section 5.4)
2. The writing needs improvement as it is currently a bit ambiguous at certain places. (See 1. and 2. In questions below for further details). For instance, plotting details for Figure 3 are unclear from the appendix. Similarly, it is unclear to me how $var(x_{t-1})$ is computed in line 9 of Algorithm 3. The overall clarity of this paper will greatly improve if these paragraphs are rewritten by adding additional details. Similarly, mathematical expressions should be added at multiple places along with text for improved clarity. For instance, Figure 2, writing $var(x_t)$ is more informative than simply writing $x_t$ for the label of y-axis. In Figure 3, the mathematical form of error can be included instead of labelling y-axis as error.
3. Certain parts of derivation of proof of Theorem 3.1 need further explanation. The proof assumes that for an image P, and pixels $p_i, p_j \in P$, $p_i \perp p_j$ if $i \neq j$ which is usually not true in practice as neighboring pixels in image usually have high correlation. It also assumes that each pixel in image $P$ follows distribution $\mathcal{N}(\mu_i, \sigma)$, but later it claims that $\mu_i$ is the mean of the distribution of $\hat{x}_{t-1}$ which looks incorrect as the mean of the latter distribution is a vector/tensor (as it is an image) while $\mu_i$ is a scalar as it is mean of pixel values.

**Questions:**

1. The explanation of details for plotting Figure 3 is unclear from the description given in Appendix B. Perhaps writing mathematical equations might make the idea more concise and clear. Also, for the purpose of rebuttal, could the authors include this expression here, or alternately provide an explanation of what the figure indicates? It is unclear to me why the error computation for this figure is split into two different stages. Any intuition/reasoning behind choosing the methodology of computing errors in Figure 3 is appreciated.
2. In DDPM sampling, we use line 4 in algorithm 2 to get the next sample $x_{t-1}$.  It is unclear to me how this $x_{t-1}$ is used to compute $var(x_{t-1})$ in line 9 of Algorithm 3. We cannot use the analytic closed form of variance as it won’t have errors/exposure bias from prediction in network. Thus it needs to be sample variance. In that case, to compute $var(x_{t-1})$, is $z \sim N(0, I)$ sampled multiple times and then sample $var(x_{t-1})$ computed? As other terms for sampling $x_{t-1}$ are fixed for a given $x_t$ (Line 4 in Algorithm 2), isn’t $z$ the only source of variance in this case? How many samples of $x_{t-1}$ are needed to get a reasonable estimate of this sample variance? Is it possible to add these details in the text that explains the algorithm?

---

> ### Author Response · Authors · 2023-11-18
> **Response to Reviewer YoEn (part1)**
>
> Thank you for the insightful questions and comments. We summarize your questions and remarks and respond below (part1):
>
> > **[W1] minor overhead in sampling time; effect of window sizes and cutoff values**
>
> * When working with a small DDPM backbone, minor overhead in sampling time is observed (Figure 5).
> However, in practice, we believe it is acceptable, for example, when we are sampling with 20 steps, it only requires less than 10ms additional sampling time on a RTX3090.
> Furthermore, for the ADM backbone, which is bigger in model size as compared to DDPM, our *TS*-DDIM requires nearly identical sampling time as DDIM (Figure 6) but achieves significantly better performance (Table 2).
> It makes our method stand out in nearly a “free lunch” situation, where we **make significant improvements in performance without sacrificing efficiency**.
> In fact, the additional computation we introduce to the sampling method only involves (1) computation of the sample variance (2) calculating $\mathop{\arg\min}(\cdot)$ within a small window size of timesteps. These two computations can be quickly achieved in modern GPUs.
>
>
> * For a larger number of sampling steps, take 100 steps for example, we only present the window sizes of {2,4,6,8} to showcase the possible effects of different situations. In fact, setting window size=8 is illogical, given 100 steps leads to step size of 10. As we mentioned in Section 5.4, since per-step estimation is much more accurate with 100 steps as compared to fewer steps, one should always stick to a small window size (e.g. 2,4), not a large window size of 8 which is almost the same as the step size.
>
> * For a smaller number of sampling steps (e.g. 10 or 20), the cutoff value can have a slightly bigger impact on the performance. However, we would like to stress that (1) One can have a good estimation of the cutoff values by plotting a plot like Figure 2. We obtain similar plots for varying datasets and models, and conclude that optimal cutoff value usually lies in the range of [200,400]. In our experiments, even under different settings (e.g. models, datasets, number of steps), in less than 5% of the cases the optimal cutoff value falls outside the range of [200,400]. This can serve as a guideline for future works.
> (2) To better showcase the problem, we are not using the optimal combination of window size and cutoff value in Figure 7. Instead, we present arbitrarily selected values. If one follows our guidance in selecting window sizes and cutoff values, one would find that the results to be less sensitive.
>
>
> * Overall, we believe that if the window size and cutoff value are wrongly estimated (e.g. window size=8 for 100 steps with step size=10), one would obtain worse performance. As the pioneering work in the field, our work provides the valuable empirical findings which make it easier to apply our method in practice.
> As stated in Section 5.4, and explained above, we provide the guidelines for selecting window sizes (large window sizes (e.g. 40, 80) for fewer steps(e.g. 5,10 steps); very small window size (e.g. 2) for more steps (e.g.50,100 steps)) and cutoff values (within the range of [200,400]).
> The findings hold true for our experiments under various settings and can serve as the guidance for applying our method in practice.
>
> ***
>
> > **[W2] suggestions on added details in paper**
>
> Thank you for the suggestions. We’ve updated the manuscript as follows: (1) changed y-axis to $var(x_t)$ in Figure 2; (2) added the details for plotting Figure 3 in Appendix B, including a visualized example presented in Figure 8; (3) added details in computing $var(x_{t-1})$ of Algorithm 3 in Appendix B. Please also refer to the comments we made below for Q1 & Q2.

---

> ### Author Response · Authors · 2023-11-18
> **Response to Reviewer YoEn (part2)**
>
> Please find part2 of our response below:
>
> > **[W3] explanation on independence assumption in derivation**
>
> Thanks for raising this insightful question. Indeed, the assumption of independence is not true for natural  images.  However, considering the image generation process of the diffusion model, we start with Gaussian distribution and gradually remove the noise to generate images. Initially, the assumption of independence is valid, since we start with pure Gaussian noise. Yet, as time approaches zero, the pixels are more and more dependent on each other, making the assumption of independence invalid. Thus our derivation is applicable when $t$ is large. This also explains the introduction of  the cutoff mechanism, where we avoid applying the time shift when $t$ is smaller than a certain predefined value.
>
> We could provide a more general derivation that takes into account the correlation between pixels; however, we would still assume that this correlation can be ignored when $t$ is large.
>
> To construct a more general derivation, one only needs to make the following modifications to our original derivation. The independence condition on $p_i$ and $p_j$ is only used in Equation 16. When assuming a general Gaussian distribution $\mathcal{N}(\mu, \Sigma)$ on the image, we need to replace the $\mathbb{E}(p_i)\mathbb{E}(p_j)$ term by  $\mathbb{E}(p_i p_j)$. Equivalently we can keep the first equation unchanged while adding an extra $\sum_{i \neq j}\mathbb{E}(p_i p_j) - \mathbb{E}(p_i)\mathbb{E}(p_j)$, which gives the covariance between $p_i$ and $p_j$, denoted by $\Sigma_{\{i,j\}}$ Therefore, Equations 16 to 19 have an additional $\sum_{i \neq j}\frac{\Sigma_{\{i,j\}}}{d(d-1)}$ at the end. Equation 20 has an additional $\sum_{i \neq j}\frac{\Sigma_{\{i,j\}}}{d-1}$ at the end and Equation 21 has an additional $-\sum_{i \neq j}\frac{\Sigma_{\{i,j\}}}{d-1}$  on the right hand side.
>
> Thanks for bringing up the ambiguity of $\mu$.  $\mu_i$ is the mean of each pixel of the $x_{t-1}$. Thus it is also a scalar.  We have corrected this in the revision.
>
> ***
>
> > **[Q1] details and intuition for plotting Figure 3**
>
> Thanks for raising this important question. Figure 3 shows the mean square error (MSE) between the prediction and ground truth at each step in the backward process. Given an image denoted as $x_0$, by applying Equation 2 ($x_t=\sqrt{\bar{\alpha}_t} x_0 + \sqrt{1-\bar{\alpha}_t}\epsilon_t$)  we could obtain a sequence of $x_t, t=1,2,\cdots, T-1$.  Taking each $x_t$ and the paired time step $t$ to run the backward process, we obtain a sequence of predicted $\hat{x}_0^t, t=1,2,\cdots, T-1$. Ideally, we would expect all these predicted $\hat{x}_0^{t}, t=1,2,\cdots, T-1$ to be exactly equal to the ground truth $x_0$, as they are generated using the given $x_0$. However, this is not the case in practice. In our experiments, we found that only when $t<t_s$ (around 650 steps in our experiment using DDIM) we could obtain the original $x_0$ by running the backward process with paired $(x_t,t)$. For $t>t_s$, the image created using $(x_t, t)$ differs from the original $x_0$. This observation also reveals that the image generation process of diffusion models basically contains two stages. In the first stage, the model moves the Gaussian distribution towards the image distribution and no modes are presented at this stage, which means we can not know which images will be generated. In the second stage, the prediction shows clear patterns and modes are presented. We can predict which images will be generated following the backward process. This observation led us to divide the error computation into two stages. The full  explanation, including the equations, are added to Appendix B with Figure 8 showing the visualized procedure.
>
> ***
>
> > **[Q2] details for computing variance in Algorithm 3**
>
> Thank you for the insightful question. Ideally, as you suggested, to accurately compute the $var(x_{t-1})$, one needs to generate many samples of $x_{t-1}$. Though it could obtain more accurate estimation of $var(x_{t-1})$, it also significantly increases the computational workload. Surprisingly, we find that under some assumption (see derivation of Theorem 3.1) the $var(x_{t-1})$ could be estimated using the inter variance of a single $x_{t-1}$, Thus during sampling, we compute the variance within each $x_{t-1}$. So there is no need to use multiple samples for estimation. Given the space limitations of the main text, we add the above details in Appendix C.

---

> ### Comment · Reviewer_YoEn · 2023-11-21
>
> Dear authors,
>
> Thank you for your detailed response. I am recalibrating my score for now on the basis of above response. I am not convinced that the approximation to replace variance of $x_{t-1}$ with variance within a single $x_{t-1}$ is completely principled.
>
> The paper uses statements like "we theoretically and empirically show that by adjusting the next time step t − 1 during sampling according to the variance of the current generated samples, one can effectively alleviate the exposure bias." However, it seems like the variance of the current generated samples is not actually computed the way it should be. A significant amount of discussion and empirical analysis in section 2.2 is also dedicated to sample variance. The paragraph following Theorem 3.1 states "Secondly, we establish the relationship between the covariance matrix of the $\hat{x}\_{t−1}$ distribution and the variance of the predicted $\hat{x}\_{t−1}$". I think these sentences need to be revised to indicate that variance of $\hat{x}\_{t-1}$ is not actually calculated but rather variance within pixels of a single $\hat{x}\_{t-1}$ is used. Again, I fail to see why this is a principled approach. Also, as computation of $t_s$ is dependent on variance of $x_{t-1}$, I'm not sure if the optimal $t_s$ is being computed at each step of Algorithm 3.

---

> ### Author Response · Authors · 2023-11-21
> **Response to further comments (Part 1)**
>
> Dear reviewer,
>
> Thanks for your feedback. In our humble opinion, there might be some misunderstandings to our method. As we discussed in the reply to [Q2], accurately computing the variance of $\hat x_{t-1}$ is practically infeasible.  Thus we proposed our method to approximate this variance using the variance within a single $\hat x_{t-1}$. The construction of our method is based on rigorous mathematical derivation, thus we think it is a principle approach. We would like to provide a more detailed explanation of our derivation below, and we hope this would resolve your doubts about our method.
>
> To better explain this, we would like to refer you to the Equation (17) in Appendix J.1, which is the key to understanding this approximation.  Equation (17) gives the rigorous mathematical derivation for the connection between the variance of the pixels (denoted by $\sigma_{t-1}$ in Equation (15)) and the variance of the predicted sample (denoted by $\sigma$).  Moreover, we can provide an intuitive explanation of why it holds based on the well-known “law of total variance” (https://en.wikipedia.org/wiki/Law_of_total_variance), which states that for two random variables $Y$ and $X$ the total variance can be splitted into $Var(Y) = E[Var(Y|X)]+Var(E(Y|X))$. In the following, we will explain how to make $Var(Y)$ correspond to our $E(\sigma_t)$, $E[Var(Y|X)]$ to $\sigma$ and $Var(E(Y|X))$ to the variance of $\mu_i$ as derived in Equation (17). We let $X$ be a uniform distribution which samples the position of the pixel, and $Y|X$ is the (random) value of the pixel on that position which follows $N(\mu_x,\sigma)$. Therefore, to evaluate $Var(Y)$, we can first sample a bunch of positions, find their values, and then compute the variance of these values. Without loss of generality, we can simply calculate the variance of all the pixels as if every position is sampled once. Thus $Var(Y)$ can be efficiently estimated by our $E(\sigma_{t-1})$ in Equation (17). For the right hand side, by our construction, $E(Var(Y|X))$ is nothing but $E(\sigma)=\sigma$ as $Y|X\sim N(\mu_x,\sigma)$. Finally, since $E(Y|X)$ is the mean of the pixel ($\mu_x$ in this illustration and $\mu_i$ in the paper), $Var(E(Y|X))$ can be estimated unbiasedly by the variance of the means (last term of Equation (17)).  In the paragraph under Equation (17),  we detailed the reasoning process from Equation (17) to Equation (19).  Through our derivation, when $t$ is large the variance of predicted $\hat x_{t-1}$ could be approximated by the variance within a single $\hat{x}_{t-1}$. Thus our method is a principle approach.
>
> Kindly let us know if you still have doubts about our method, we will try our best to explain it better.

---

> ### Author Response · Authors · 2023-11-21
> **Response to further comments (Part 2)**
>
> Regarding the sentences you mentioned, we stated in Appendix B that we used the variance within $\hat x_{t-1}$ in Section 2.2. As for the statement in the introduction "we theoretically and empirically show that by adjusting the next time step t − 1 during sampling according to the variance of the current generated samples, one can effectively alleviate the exposure bias.",  though we did not accurately compute the variance of the generated samples, we did approximate it using Theorem 3.1. We agree that clearly stating we use the approximation variance of the generated sample will make our paper clearer. For the other statement following Theorem 3.1, we use covariance matrix and variance to represent the true variance of $\hat x_{t-1}$ and pixel variance within a single $\hat x_{t-1}$, respectively.  We agree that this might confuse the readers, and we will revise it in the manuscript.

---

> > ### Comment · Reviewer_YoEn · 2023-11-22
> >
> > Dear authors,
> >
> > Thank you for your response. I understand the math behind Eq 17-19. An issue with the current draft is that many assumptions made within the proof are not stated clearly in the main paper. Ideally, a reader doesn't have to read the proof/appendix to understand the assumptions under which the theorem holds.
> >
> > I am revising my score but I would highly suggest the authors to rewrite the theorem statement in a more rigorous manner as the derived expression for variance only holds under the assumption of independence of pixels and for large $t$. The theorem statement should clearly state all these assumptions.
> >
> > I also think that the sentence in the revised draft is ambiguous "Secondly, we establish the relationship between the variance within the samples of $\hat{x}\_{t−1}$ and the variance of $\hat{x}\_{t−1}$."

---

> > > ### Author Response · Authors · 2023-11-22
> > >
> > > Dear reviewer,
> > >
> > > Thank you very much for your constructive comments and suggestions.
> > >
> > > We’ve added the assumptions that you are referring to in Theorem 3.1 of the main text of the paper.
> > > Specifically, Theorem 3.1 holds under these two assumptions:
> > >
> > > (1) If $t_s$ is close to $t-1$, then $\sqrt{\bar \alpha_{t-1}} - \sqrt{\bar \alpha_{t_s}} \approx 0$.
> > >
> > > (2) When $t-1$ is large, the $\hat  x_t$ is still close to the initialized normal distribution, then we assume the pixels in $\hat x_t$ are independent of each other (i.e. the distribution has a diagonal covariance matrix).
> > >
> > > Now the revised Theorem 3.1 incorporates these two assumptions.
> > >
> > > As stated in Appendix J.1: The first assumption corresponds to the selection window we are using, which guarantees that $t_s$ is close to $t-1$. The second assumption corresponds to the cutoff mechanism, which guarantees that we only do time shifts when $t$ is large.
> > >
> > > Regarding the sentence added in the revised draft, indeed it is ambiguous. We have changed it to: “Secondly, we establish the relationship between the variance within a single sample of $\hat x_{t-1}$ and the variance of $\hat x_{t-1}$”, where the latter refers to the variance computed by considering the elements with the same index across all samples of $\hat x_{t-1}$.

---

### Official Review · Reviewer_8DgG · 2023-11-05

**Soundness:** 3 good
**Presentation:** 3 good
**Contribution:** 3 good
**Rating:** 6
**Confidence:** 4

**Summary:**

The paper proposes to select a suitable timestep for next step instead decreasing it as in most diffusion model sampling algorithms.

**Strengths:**

1. The paper is well-written and organized.

2. The analysis are sufficient and the findings in section 3. are interesting.

3. According to the tables, the FIDs are improved with different baseline samplers. It is also good to see that the sampling time is not increased significantly as shown in figure 6.

**Weaknesses:**

1.  Other metric should also be included, such as precision and recall. Is there any reason why text-to-image performances are not added? It would be better if authors could include such results as the task is one of most important tasks of diffusion model.

2. Could you also visualize the selecte timestep trajectory?  It also would better to have more analysis on the selected timesteps and around which timesteps are mostly important.

**Questions:**

as above

---

> ### Author Response · Authors · 2023-11-18
> **Response to Reviewer 8DgG**
>
> Thank you for your constructive comments and suggestions. Please find our response below:
>
> > **[W1] Other metric should also be included, such as precision and recall. Is there any reason why text-to-image performances are not added? It would be better if authors could include such results as the task is one of most important tasks of diffusion model.**
>
> We followed previous works [1,2] by reporting results in FID on the widely used benchmark datasets such as CIFAR-10, CelebA and LSUN. But, yes, reporting results on other metrics and the results in text-to-image generation tasks would definitely make our contribution stronger. Thus, we report these results as follow:
>
> | Sampling Method | 10 steps | 20 steps|
> |---|---|---|
> |DDIM | 27.80 | 25.47 |
> |*TS*-DDIM| **26.32 (+5.32%)**|**24.80 (+2.63%)**|
>
> The results are also presented in Table 10 in Appendix I of the revised manuscript.
>
> We report precision (P) and recall (R) with ADM as backbone on CIFAR10 using DDIM and our *TS*-DDIM. The results are also included in Table 7 in Appendix I of the revised manuscript.
>
> | Sampling Method | 5 steps, P | 5 steps, R | 10 steps, P | 10 steps, R| 20 steps, P | 20 steps, R | 50 steps, P | 50 steps, R |
> |---|---|---|---|---|---|---|---|---|
> |DDIM | 0.59 | 0.47 | 0.62 | 0.52 | 0.64 | 0.57 |0.66 | 0.60 |
> |*TS*-DDIM| 0.57 | 0.46 | 0.62 | 0.55 | 0.64 | 0.60 | 0.65 | 0.62 |
>
> [1] Jiaming Song, Chenlin Meng, and Stefano Ermon. Denoising diffusion implicit models. In International Conference on Learning Representations, 2021.
>
> [2] Luping Liu, Yi Ren, Zhijie Lin, and Zhou Zhao. Pseudo numerical methods for diffusion models on manifolds. In International Conference on Learning Representations, 2022.
>
> ***
>
> > **[W2] Could you also visualize the selected timestep trajectory? It also would better to have more analysis on the selected timesteps and around which timesteps are mostly important.**
>
> We visualize the selected timestep trajectory for experiments on CIFAR-10 with *TS*-DDIM using DDPM backbone and 10 sampling steps. Please refer to Figure 12 in Appendix G of the revised manuscript. The red dashed line is the original timesteps in DDIM (uniform selection). The blue line shows the trajectories using *TS*-DDIM. For better visibility, we also zoom in for the range of [300,600]. In the range of (300, 600), more time shifts happen, and time steps can shift to both larger or smaller time steps within the window. Figure 12 demonstrates that most of the time shifting happens in the intermediate range of the backward process. This phenomenon might be due to the fact that at the initial stage (when $t$ is large), the samples are predominantly influenced by the initialized normal distribution, leading to minimal variance change. Conversely, in the later stage (when $t$ is small), the samples are predominantly composed of image data, containing most of the image information. This makes it easier for the model to make accurate predictions.

---

### Author Response · Authors · 2023-11-18
**General response for all reviewers**

Dear reviewers,

We sincerely appreciate your insightful comments and the dedicated time you've invested in reviewing our paper.
We are encouraged to hear that our paper is well-written and organized (Reviewer 8DgG) and our work presents interesting findings (Reviewer 8DgG, Reviewer zG28) and valuable contributions (Reviewer YoEn).
We are also delighted to know that all reviewers acknowledge the consistent improvements we made over previous methods without sacrificing sampling efficiency (Reviewer 8DgG) or retraining the model (Reviewer YoEn, Reviewer zG28).
We would also like to thank the reviewers for suggesting additional experiments. The revisions we made to the paper are marked in red, which we summarize below:

* We updated the labels of the y-axis in Figures 2&3, and added more details on how to plot Figure 3 in Appendix B.
* We presented an example of selected timestep trajectory in Figure 12 in Appendix G. We also add more details in computing the variance in Algorithm 3 in Appendix C.
* We added additional results in Appendix I, including experiments with DPM-solver, DEIS, text-to-image generation and ImageNet.
* We added details on deriving an analytical estimation of the window size in Appendix J.2.

---

### Author Response · Authors · 2023-11-21

Dear reviewers,

Thank you all again for the valuable work you have done and the insightful feedback you have provided on our paper. We are interested in knowing whether our answers have clarified your doubts or if there are still aspects that remain unclear. Should you have further questions on any topic, even those not previously discussed, we would be more than willing to continue the discussion and provide additional details.

Best regards,

The authors

---

### Meta-Review · Area_Chair_ASre · 2023-12-10

**Metareview:**

Four knowledgeable referees reviewed this submission. The reviewers raised concerns w.r.t.:
1. Missing metrics such as precision / recall (8DgG)
2. Missing analysis of the importance of different time-steps (8DgG)
3. The presentation of the paper, which could be significantly improved  - e.g. by adding explanations on the derivation of the presented proofs - (YoEn)
4. The claims made w.r.t. the contribution, as the contribution appeared more on the empirical side (YoEn, zG28)
5. The unclear performance of the proposed approach when combined with alternative solvers (zG28)
6. The justification and sensitiveness to hyper-parameter choices (rQd1,YoEn)
7. The overhead in sampling time (zG28, YoEn).

The rebuttal adequately addressed the reviewers' concerns by providing additional results and metrics, by clarifying the theoretical and empirical contribution, and by discussing the sampling time and the effect of hyper-parameters. During discussion, the reviewers acknowledge the authors' efforts to address their concerns. The reviewers are overall convinced and unanimously recommend to accept. The AC agrees with the reviewers assessment and recommends to accept.

**Justification For Why Not Higher Score:**

The experimental evidence is based on low resolution data. The contribution could strengthened by including experiments with alternate solvers on higher resolution and larger scale datasets.

**Justification For Why Not Lower Score:**

The initial concerns raised by the reviewers were addressed during the rebuttal and discussion phases. Reviewers agree that the proposed approach appears promising and could be interesting to the ICLR community, and unanimously recommend to accept.

---

### Decision · Program_Chairs · 2024-01-16

Accept (poster)